# Block copolymer derived uniform mesopores enable ultrafast electron and ion transport at high mass loadings

Tianyu Liu [1], Zhengping Zhou[1], Yichen Guo[1], Dong Guo[1] & Guoliang Liu [1,2,3]

High mass loading and fast charge transport are two crucial but often mutually exclusive characteristics of pseudocapacitors. On conventional carbon supports, high mass loadings inevitably lead to sluggish electron conduction and ion diffusion due to thick pseudocapacitive layers and clogged pores. Here we present a design principle of carbon supports, utilizing self-assembly and microphase-separation of block copolymers. We synthesize porous carbon fibers (PCFs) with uniform mesopores of 11.7 nm, which are partially filled with $MnO_2$ of <2 nm in thickness. The uniform mesopores and ultrathin $MnO_2$ enable fast electron/ion transport comparable to electrical-double-layer-capacitive carbons. At mass loadings approaching 7 mg cm$^{-2}$, the gravimetric and areal capacitances of $MnO_2$ (~50% of total mass) reach 1148 F g$^{-1}$ and 3141 mF cm$^{-2}$, respectively. Our $MnO_2$-coated PCFs outperform other $MnO_2$-based electrodes at similar loadings, highlighting the great promise of block copolymers for designing PCF supports for electrochemical applications.

[1] Department of Chemistry, Virginia Tech, Blacksburg, VA 24061, USA. [2] Macromolecules Innovation Institute, Virginia Tech, Blacksburg, VA 24061 USA. [3] Division of Nanoscience, Virginia Tech, Blacksburg, VA 24061, USA. Correspondence and requests for materials should be addressed to G.L. (email: gliu1@vt.edu)

High mass loading and fast charge transport are at the heart of electrochemical energy storage[1–3]. The former is crucial for high energy per device, and the latter for high power[4]. Low-cost, high-capacitance, and environment-benign pseudocapacitive $MnO_2$ are loaded on electrically conductive supports and used as supercapacitor electrodes with a theoretical limit of 1367 F g$^{-1}$ (based on a potential window of 0.8 V)[5–9]. Toward commercialization, the mass loading of the total active materials must be at least 5 mg cm$^{-2}$[10]. However, high mass loadings often lead to thick and dense layers of insulating $MnO_2$ ($10^{-5}$~$10^{-6}$ S cm$^{-1}$) on the supports[11–14]. Consequently, the internal resistance increases and the ion diffusion is perturbed, resulting in sluggish charge transport—both electron conduction and ion diffusion[5,11,15]. Here we innovate the design of porous carbon fiber (PCF) as a lightweight, flexible, binder-free, and conductive-additive-free support for $MnO_2$. Using the disparate concept of block copolymer microphase-separation to generate uniform mesopores in PCFs, we have bridged the two mutually exclusive characteristics, i.e., high mass loadings and ultrafast electron/ion transport.

An ideal support for $MnO_2$ and other transition metal oxides ($RuO_2$, NiO, $WO_3$, and $Fe_2O_3$, etc.) needs the characteristics of (1) lightweight, (2) large surface areas for high loadings, (3) high electron conductivity, and (4) low ion diffusion resistivity. However, there is not a single nanostructure that meets all these characteristics[5,15]. Carbon supports are inherently lightweight and electrically conductive. At high mass loadings of transition metal oxides, the electrical conductivity of electrodes decreases, but it can be restored by blending or wrapping with additional conjugated polymers[16,17] or carbon additives[16,18,19], as shown for excellent supports such as wearable textile structures[16] and graphene[16,20]. The ion conduction, however, is drastically complicated[21], and the efficient ion diffusion across the entire support, as well as the thick layer of $MnO_2$, remains a significant challenge. To mitigate the ion diffusion resistivity, ultrathin layers of $MnO_2$ have been deposited on model supports, e.g., nanoporous Au[22,23], Pt foil[9], Ni foil[24], Si wafer[25], dendritic Ni[26] and macroporous Ni film[27]. With a thickness of <10 nm[22] or at a mass loading of <0.35 mg cm$^{-2}$ on the model supports[23,26], $MnO_2$ exhibits fast electron/ion transport and the gravimetric capacitances approach the theoretical limit. Nevertheless, when the conventional lightweight carbon supports are loaded with $MnO_2$, they either suffer from a limited surface area for depositing a large amount of $MnO_2$ thin layers (e.g., carbon cloth[11,12], carbon fibers[16,28–30] and other macroporous carbons[13,14]), or they lack desirable porous structures that facilitate rapid ion diffusion across long distances to maintain high rate capability (e.g., microporous carbons[5,15,31]).

Ultimately the key to high mass loading and fast electron/ion transport lies at the design of porous carbon architectures[32,33]. We hypothesize that mesoporous carbon fibers with a narrow pore size distribution are the most preferable for addressing the challenges of high mass loading and fast electron/ion transport. Conversely, micropores are susceptible to clogging after loading with $MnO_2$ and thus provide sluggish ion transport, while macropores offer limited surface areas for high mass loadings of transition metal oxides. In addition, non-uniform mesopores lead to inefficient use of the surface area for depositing $MnO_2$ and potential clogging of the small pores.

To test our hypothesis, herein we demonstrate block copolymer-derived PCFs as lightweight and high mass-loading supports for $MnO_2$ (Fig. 1). Because block copolymers self-assemble and microphase separate into uniform and continuous nanoscale domains[34–41], after pyrolysis they generate interconnected mesoporous carbons with large surface areas for depositing $MnO_2$. Disparate from all other carbon supports, the mesopores are designed from the macromolecular level and offer a high degree of uniformity. Importantly, our judiciously designed mesopores have an average diameter of 11.7 nm and are partially filled with a <2-nm-thick layer of $MnO_2$ (Fig. 1c). On the one hand, the remaining mesopores provide continuous channels for efficient ion transport across the entire electrode, significantly reducing the ion diffusion resistance. On the other hand, the fibrous carbon network provides expressways for efficient electron transport without the need for any conductive additives. This contrasts with hard-templated mesoporous carbon particulates (e.g., CMK-3[31]), which demand polymer binders to hold the discrete carbon particulates together. At high mass loadings approaching 7 mg cm$^{-2}$, the PCF-supported $MnO_2$ electrodes (PCF@$MnO_2$) show superior electron/ion transport and outstanding charge-storage performances.

## Results

**Morphology.** To illustrate the crucial importance of uniform mesopores for high mass loading of $MnO_2$, we have synthesized two types of carbon fibers, i.e., PCFs with uniform mesopores derived from poly(acrylonitrile-*block*-methyl methacrylate) (PAN-*b*-PMMA) and conventional carbon fibers (CFs) with limited mesopores from pure polyacrylonitrile (PAN). Scanning electron microscopy (SEM) shows the contrasting morphologies of PCFs and CFs (Fig. 2a, d and Supplementary Fig. 1). Owing to the microphase separation of PAN-*b*-PMMA and the subsequent degradation of poly(methyl methacrylate) (PMMA), the PCFs were perforated with a large number of uniformly distributed, randomly oriented, and interconnected mesopores of ~11.7 nm

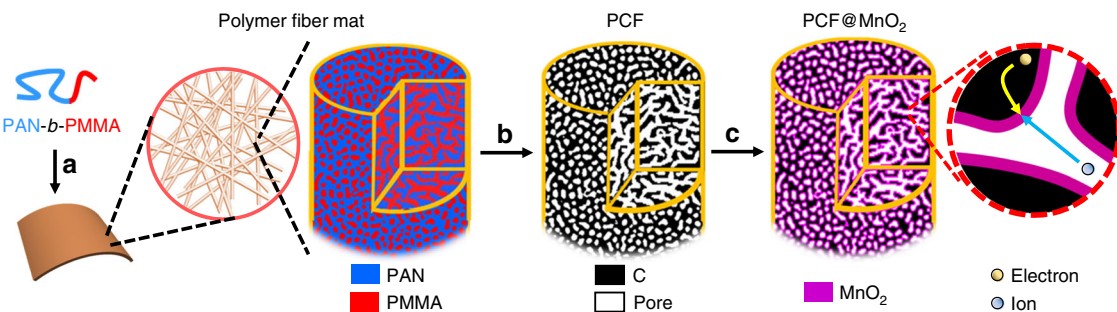

**Fig. 1** Schematic illustration of the synthesis of PCF and PCF@$MnO_2$. **a** PAN-*b*-PMMA block copolymer is spun into a polymer fiber mat. (Magnified view) PAN (blue) and PMMA (red) in the block copolymer fibers microphase separate into a bicontinuous network structure. Via thermal oxidation, PAN is crosslinked to ensure a high yield of conductive carbon network. **b** Upon pyrolysis, the polymer fibers are converted to porous carbon fibers (PCFs, black) with continuous and uniform mesopores (white channels), which afford high loadings of transition metal oxides. **c** The PCFs are loaded with $MnO_2$ (magenta) to become PCF@$MnO_2$ through a solution-based redox deposition reaction. (Magnified view) The continuous carbon fiber matrix and the partially filled mesoporous channels provide effective expressways for electron conduction and ion diffusion, respectively

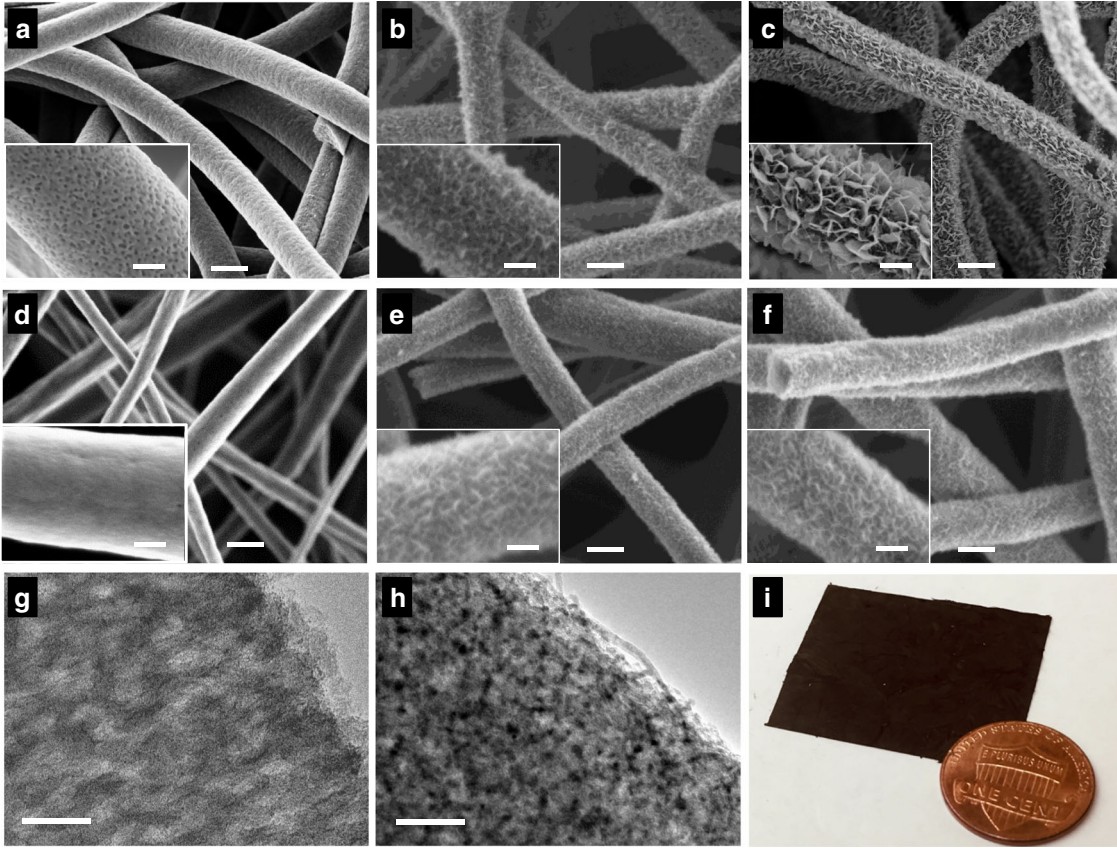

**Fig. 2** Morphology characterizations. **a–f** SEM images of (**a**) PCF, (**b**) PCF@MnO₂-1h, (**c**) PCF@MnO₂-2h, (**d**) conventional CF, (**e**) CF@MnO₂-1h, and (**f**) CF@MnO₂-2h. Scale bars: 200 nm. (Insets) Magnified views of the single fibers. Scale bars: 50 nm. Due to the surface effect, the interconnected mesopores in PCFs appear as discrete dark spots on the fiber skin (inset a) but are absent in the conventional CFs (inset d). **g, h** TEM images of PCF (**g**) before and (**h**) after loading MnO₂ for 2 h. Scale bars: 50 nm. The MnO₂ nanosheets are removed from the fiber skin to reveal the internal structures without interference. **g** The gray and dark areas correspond to the interconnected mesopores and the carbon matrix, respectively. **h** MnO₂ appear darker than carbon due to the higher atomic number. **i** A photograph of a piece of PCF@MnO₂-2h electrode next to a U.S. penny with a diameter of ~1.9 cm

(Figs. 2a, 3a, and Supplementary Fig. 1a)[42]. In contrast, the CFs derived from PAN exhibited relatively smooth surfaces and no observable mesopores under SEM (Fig. 2d and Supplementary Fig. 1b). Small angle X-ray scattering (SAXS) spectroscopy confirmed the microphase separation of PAN-$b$-PMMA and revealed that the average center-to-center pore-spacing in PCFs was 25.7 nm (Supplementary Fig. 2). The volume fraction of PAN in PAN-$b$-PMMA was ~65%, and supposedly the block copolymer should self-assemble into either cylindrical or gyroidal structures, depending on the incompatibility of the two blocks. After pyrolysis, however, the porous carbon fibers showed no well-defined cylindrical or gyroidal structures but interconnected mesopores that were irregularly shaped and uniformly distributed, as shown in the cross-sectional SEM image (Supplementary Fig. 1a). This morphology is attributed to the crosslinking of PAN at elevated temperatures, which hindered the microphase separation of PAN-$b$-PMMA into well-defined cylindrical or gyroidal structures, similar to the crosslinking-induced hindering effect in previous reports[43,44].

The two types of carbon fibers were immersed in aqueous solutions of potassium permanganate (KMnO₄, 10 mM) at 80 °C to deposit MnO₂ on their surfaces. We chose the solution-based redox deposition because it creates a conformal and homogenous layer of MnO₂ inside the pores via a self-limiting redox reaction between KMnO₄ and carbon[32,45,46]. Compared with electrochemical deposition (Supplementary Fig. 3), the redox reaction deposition is advantageous because it yields uniform and homogeneous layers of MnO₂ on PCFs that ensure a low ion

diffusion resistance and thus, a high rate capability. After the deposition, the carbon fibers were washed thoroughly with deionized water, and the supernatant were analyzed with UV-vis spectroscopy to assure that there was no residual KMnO₄ in the carbon fibers (Supplementary Fig. 4). As shown by SEM, MnO₂ started to grow confocally on PCF within the first hour (Fig. 2b), and it continued to grow into nanosheets when the deposition time was prolonged to 2 h (Fig. 2c). The growth of MnO₂ on conventional CFs, however, differed drastically. After depositing for 2 h, the surface of CF@MnO₂-2h (Fig. 2f) did not change significantly from CF@MnO₂-1h (Fig. 2e). Only a thin layer of MnO₂ nanosheets was present on the surfaces of both CF@MnO₂-1h and CF@MnO₂-2h, confirming that the block copolymer-derived PCFs afford a much higher loading of MnO₂ than pure PAN-derived CFs. To verify the successful deposition of MnO₂ in the mesopores, we compared the transmission electron microscopy (TEM) images of PCF (Fig. 2g) and PCF@MnO₂-1h (Fig. 2h). Black spots of MnO₂ were uniformly embedded in PCF@MnO₂-1h, while they were absent in PCF before loading with MnO₂. MnO₂ appeared black because Mn has a higher atomic number than carbon does. The PCF mats were prepared on a large scale and ready for use as electrodes without binders or conductive additives (Fig. 2i).

**Chemical and physical properties.** X-ray photoelectron spectroscopy (XPS), Raman spectroscopy, and high-resolution TEM orthogonally verified the successful loading of MnO₂ onto PCF.

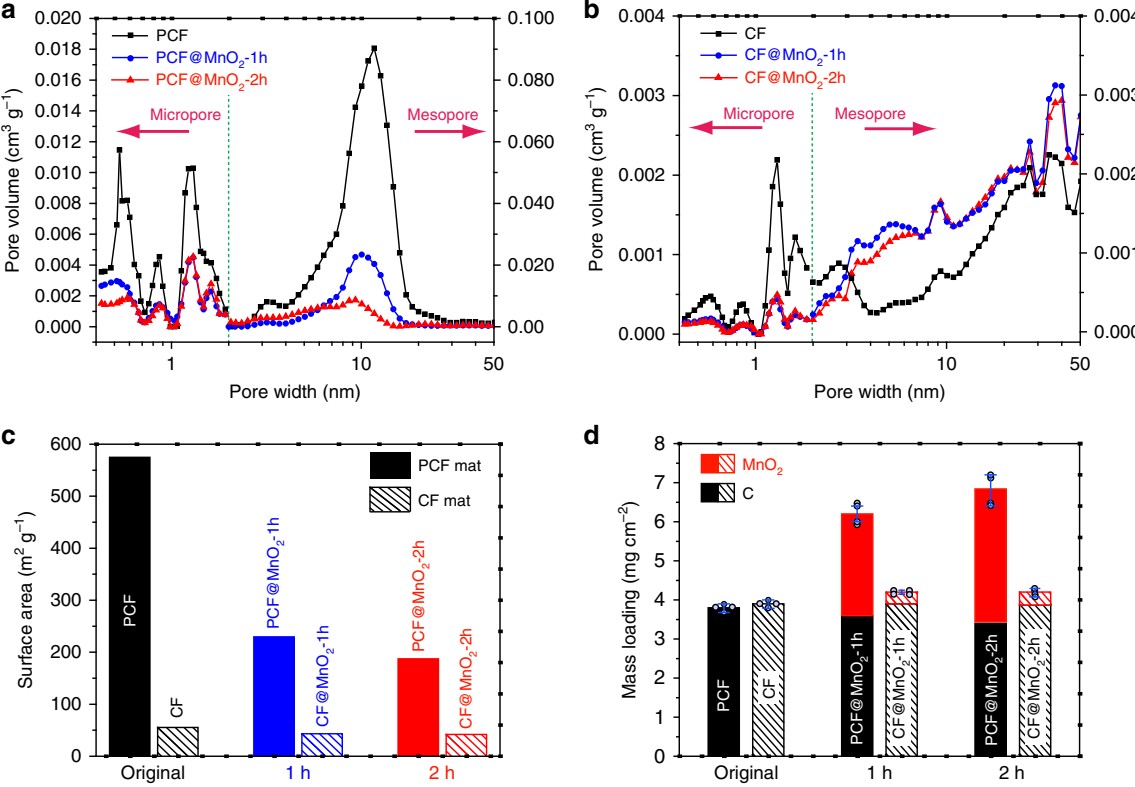

**Fig. 3** Physical characterizations. **a**, **b** Pore size distributions of (a) PCF, PCF@MnO$_2$-1h, and PCF@MnO$_2$-2h and (b) CF, CF@MnO$_2$-1h, and CF@MnO$_2$-2h. The micropore and mesopore size distributions are measured by the physisorption of carbon dioxide (at 273 K) and nitrogen (at 77 K), respectively, and calculated using the density functional theory. Note the different scales in the micropore and mesopore ranges. Compared to PCFs, CFs contain one order of magnitude lower mesopore volume. The high mesopore volume of PCFs confirms that the mesopores are interconnected and thus are accessible to the adsorbates. **c** The surface areas of PCFs and CFs before and after loading MnO$_2$. The deposition reaction time increased from 0 to 2 h. **d** Histograms of the mass loadings of (black) carbon fibers and (red) MnO$_2$ on supercapacitor electrodes. The solid and dashed bars represent electrodes composed of PCFs and conventional CFs, respectively. The error bars in (**d**) are standard deviations determined from at least four independent measurements

The XPS spectrum (Supplementary Fig. 5a) of PCF@MnO$_2$-2h showed peaks of C, O, and N corresponding to the carbon fibers, as well as a full set of peaks corresponding to Mn. An examination of the Mn 3$s$ core-level XPS spectrum (Supplementary Fig. 5b) revealed that the separation between the doublet was 4.89 eV, corroborating the valence state of Mn(IV)[47]. After MnO$_2$ deposition, the Raman spectrum of PCF@MnO$_2$-2h (Supplementary Fig. 6a) showed a group of peaks centered at ~600 cm$^{-1}$ corresponding to birnessite-phase manganese dioxide ($\delta$-MnO$_2$)[48]. The birnessite-phase of MnO$_2$ was also proven by the characteristic lattice fringes in the lattice-resolved TEM images (Supplementary Figs. 6b, c). Among the various types of MnO$_2$, $\delta$-MnO$_2$ is one of the most suitable phases for fast charge-discharge because its layered structure allows for rapid ion diffusion[49].

The porous structures of carbon fibers changed after loading with MnO$_2$. The pore size distributions of mesopores and micropores were evaluated by nitrogen and carbon dioxide adsorption-desorption isotherms, respectively (Supplementary Fig. 7). PCFs possessed significantly larger numbers of both mesopores and micropores. After depositing MnO$_2$, the micropore volumes of PCFs and CFs steadily decreased at all pore widths, but the peak positions remained unchanged (Fig. 3a, b), suggesting that the micropores were either completely filled or clogged by MnO$_2$. The pore size distributions of PCFs and CFs, however, were different in the mesopore range. PCFs exhibited appreciable decrease in the mesopore volume after depositing MnO$_2$. In addition, the peak position shifted from 11.7 to 10.0 nm after 1 h, and further down to 9.3 nm after 2 h, suggesting that the

average thickness of the MnO$_2$ layer inside the pores was ~0.9 nm and ~1.2 nm after depositing for 1 and 2 h, respectively. These thicknesses are desirable for high capacitive performance, as suggested by the Au model in a previous report[22]. The reduction of mesopore size suggests that the mesopores were only partially filled with MnO$_2$, and therefore they remained accessible to the gas adsorbates and ions. As shown in Supplementary Table 1, the pore volume reduced more in the mesopore range (86.1% reduction after the 2-h deposition) than in the micropore range (66.0% reduction after 2-h deposition). On the contrary, the mesopore volume of CFs, which was two orders of magnitudes lower than that of PCFs, increased after depositing MnO$_2$ (Fig. 3b). The increase in the mesopore volume of CFs is ascribed to the porous structures formed by MnO$_2$ as shown in Fig. 2e, f. The total pore volumes of CF-based electrodes were at least one order of magnitude lower than those of PCF-based electrodes.

The incorporation of MnO$_2$ into PCFs and CFs also altered the surface area (Fig. 3c). The surface area of the PCF mat (574.8 m$^2$ g$^{-1}$) was more than ten times higher than that of the CF mat (55.31 m$^2$ g$^{-1}$). Upon loading with MnO$_2$, the surface area of PCFs decreased from 574.8 to 229.5 m$^2$ g$^{-1}$ for PCF@MnO$_2$-1h, and further down to 187.5 m$^2$ g$^{-1}$ for PCF@MnO$_2$-2h. In contrast, the surface area of CFs only experienced moderate decreases from 55.31 to 43.35 m$^2$ g$^{-1}$ for CF@MnO$_2$-1h and to 41.76 m$^2$ g$^{-1}$ for CF@MnO$_2$-2h.

The higher loadings of MnO$_2$ in PCFs than in CFs is due to the large number of uniform mesopores (Fig. 3d). The total mass loadings (including carbon fibers and MnO$_2$) of PCF, PCF@MnO$_2$-1h, and PCF@MnO$_2$-2h were 3.8 ± 0.1, 6.2 ± 0.3,

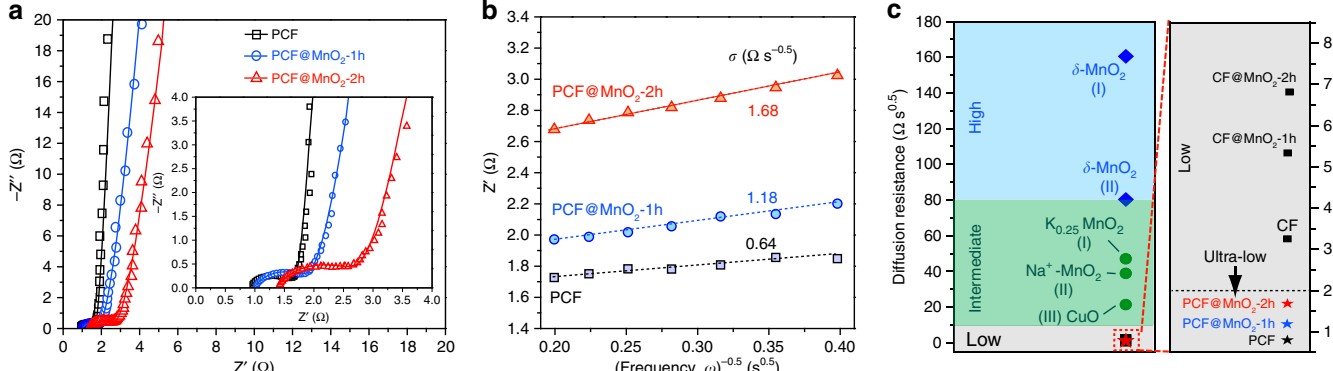

**Fig. 4** Ultra-fast electron and ion transport in the PCF-based electrodes. **a** Nyquist plots collected at open circuit potentials with 5 mV perturbation and a frequency range from 10,000 to 0.1 Hz. The open symbols are experimental data and the solid lines are fitting curves. **b** $Z'$ vs. the reciprocal of the square root of frequency ($\omega^{-0.5}$) in the intermediate frequency range. The dashed lines are best fitting lines to calculate the diffusion resistance, $\sigma$. **c** The ion diffusion resistance of the carbon fiber electrodes in comparison with other reported electrodes: (I) Ref. [57]; (II) Ref. [58]; (III) Ref. [59]. The CF-based electrodes show low diffusion resistance and the PCF-based electrodes show ultra-low diffusion resistance

and $6.8 \pm 0.4$ mg cm$^{-2}$, respectively. The error bars ($\pm$ values) are standard deviations determined from at least four independent measurements. MnO$_2$ accounted for 42% and 50% of the total mass of PCF@MnO$_2$-1h and PCF@MnO$_2$-2h, respectively. In contrast, CF-based electrodes showed much smaller mass loadings of $3.9 \pm 0.1$, $4.2 \pm 0.1$, and $4.3 \pm 0.2$ mg cm$^{-2}$ for CF, CF@MnO$_2$-1h and CF@MnO$_2$-2h, respectively. MnO$_2$ only contributed ~9% of the total mass of CF@MnO$_2$-1h and CF@MnO$_2$-2h. The difference in the mass loading of MnO$_2$ on PCFs and CFs is also apparent in the SEM images (Fig. 2). The comparison shows that the uniform mesopores are indispensable in realizing the high mass loading of MnO$_2$ on the carbon fibers. The abundant mesopores provide large solution-accessible surface areas for loading MnO$_2$ on the PCFs, while micropores can only host a limited amount of MnO$_2$ because the deposition solution can barely access them. The mass of all carbon fibers reduced slightly after loading with MnO$_2$, due to the consumption of carbon by the redox reaction between carbon and KMnO$_4$. Further elongating the deposition time to 3 h showed no appreciable increase in MnO$_2$ loading, confirming that the redox deposition was self-limited.

**Ultra-fast electron and ion transport.** Considering the high loading of MnO$_2$ and the large number of mesopores for ion transport, we investigated the performance of the PCF-based electrodes for pseudocapacitors. The electron transport and ion diffusion resistivity were analyzed with electrochemical impedance spectroscopy (EIS). The Nyquist plots of PCF, PCF@MnO$_2$-1h and PCF@MnO$_2$-2h (Fig. 4a) exhibited incomplete semicircles followed by linear tails, which resemble the features of mixed kinetic-diffusion-controlled processes and are typical for pseudocapacitive materials[50]. To obtain the resistances, we fitted the EIS spectra with an equivalent electric circuit (Supplementary Fig. 8). The combined series resistances ($R_s$) of PCF and PCF@MnO$_2$-1h were 1.0 $\Omega$, and that of PCF@MnO$_2$-2h increased to 1.4 $\Omega$ (Fig. 4a inset). The $R_s$ values were comparable to highly conductive carbon-based materials in aqueous electrolytes[51–53], indicating that MnO$_2$ introduced minimal changes to the electrical resistance of the electrodes despite the high loadings. In addition, the charge-transfer resistances ($R_{ct}$, the semicircles in Fig. 4a inset) of PCF, PCF@MnO$_2$-1h and PCF@MnO$_2$-2h are 0.74, 0.86 and 1.30 $\Omega$, respectively. The small resistances suggest efficient electron transfer associated with the redox reaction of MnO$_2$. The augmentation of charge-transfer resistance in PCF@MnO$_2$-2h is mainly due to the increased thickness of MnO$_2$

deposited in the mesopores (evidenced by the reduction in mesopore-width shown in Fig. 3d). The increased thickness elongates the electron transport distance in MnO$_2$ and therefore obstructs electron transfer at the MnO$_2$/electrolyte interface, because MnO$_2$ is a poor electron conductor ($10^{-5}$~$10^{-6}$ S cm$^{-1}$). The small $R_s$ and $R_{ct}$ are key attributes of the block copolymer-based carbon fiber electrodes because 1) unlike discrete carbon particles or graphene flakes, the carbon fibers offer continuous expressways for electron conduction, and 2) the block copolymers endow the carbon fibers with high surface areas to load with an ultrathin layer of $\delta$-MnO$_2$, which mitigates the insulating problem and facilitates the electron transport.

In addition to the efficient electron transport, the block copolymer-derived PCF electrodes exhibited ultra-fast ion diffusion kinetics, as featured by their ultra-small diffusion resistances ($\sigma$). The values of $\sigma$ were extracted from the slopes of the linear fitting lines of the real part of impedance ($Z'$) versus the reciprocal of the square root of frequency ($\omega^{-0.5}$) (Fig. 4b). PCFs displayed the smallest $\sigma$ of 0.64 $\Omega$ s$^{-0.5}$, followed by PCF@MnO$_2$-1h (1.18 $\Omega$ s$^{-0.5}$) and PCF@MnO$_2$-2h (1.68 $\Omega$ s$^{-0.5}$). The slight increase in $\sigma$ is in accordance with the fact that the pseudocapacitive reactions are slower than the adsorption-desorption of ions pertaining to the electrical double layer capacitive processes, as well as that the mesopore size is reduced. Despite the increase, the $\sigma$ values of our PCF-based electrodes were remarkably smaller than other MnO$_2$-based materials (Fig. 4c). Notably, the $\sigma$ value of PCF@MnO$_2$-2h was even lower than that of CF (Fig. 4c and Supplementary Fig. 9), a mostly electrical double layer capacitive (EDLC) material that has fast ion diffusion kinetics. In addition, the $\sigma$ value of PCF@MnO$_2$-2h ($< 2 \Omega$ s$^{-0.5}$) is ~3.5 times lower than that of CF@MnO$_2$-2h (~7 $\Omega$ s$^{-0.5}$), highlighting the critical role of the uniform distributed, randomly oriented, and interconnected mesopores in accelerating electrolyte infiltration and ion diffusion in block copolymer-derived PCFs.

**Pseudocapacitive performance.** With continuous electron conduction and ultra-low ion diffusion resistivity, PCF@MnO$_2$-2h exhibited ultra-fast charge and discharge kinetics. The cyclic voltammograms (CVs) of PCF@MnO$_2$-2h were nearly rectangular (Fig. 5a), reflecting the rapid electron and ion transport in the electrode[54]. The current density of a supercapacitor, $i$, scales with the scan rate, $v$, following the relationship of $i = kv^b$. The power-law exponent, $b$, is an important metric to evaluate the charge-storage kinetics, and $b = 1$ for an ideal supercapacitor. By

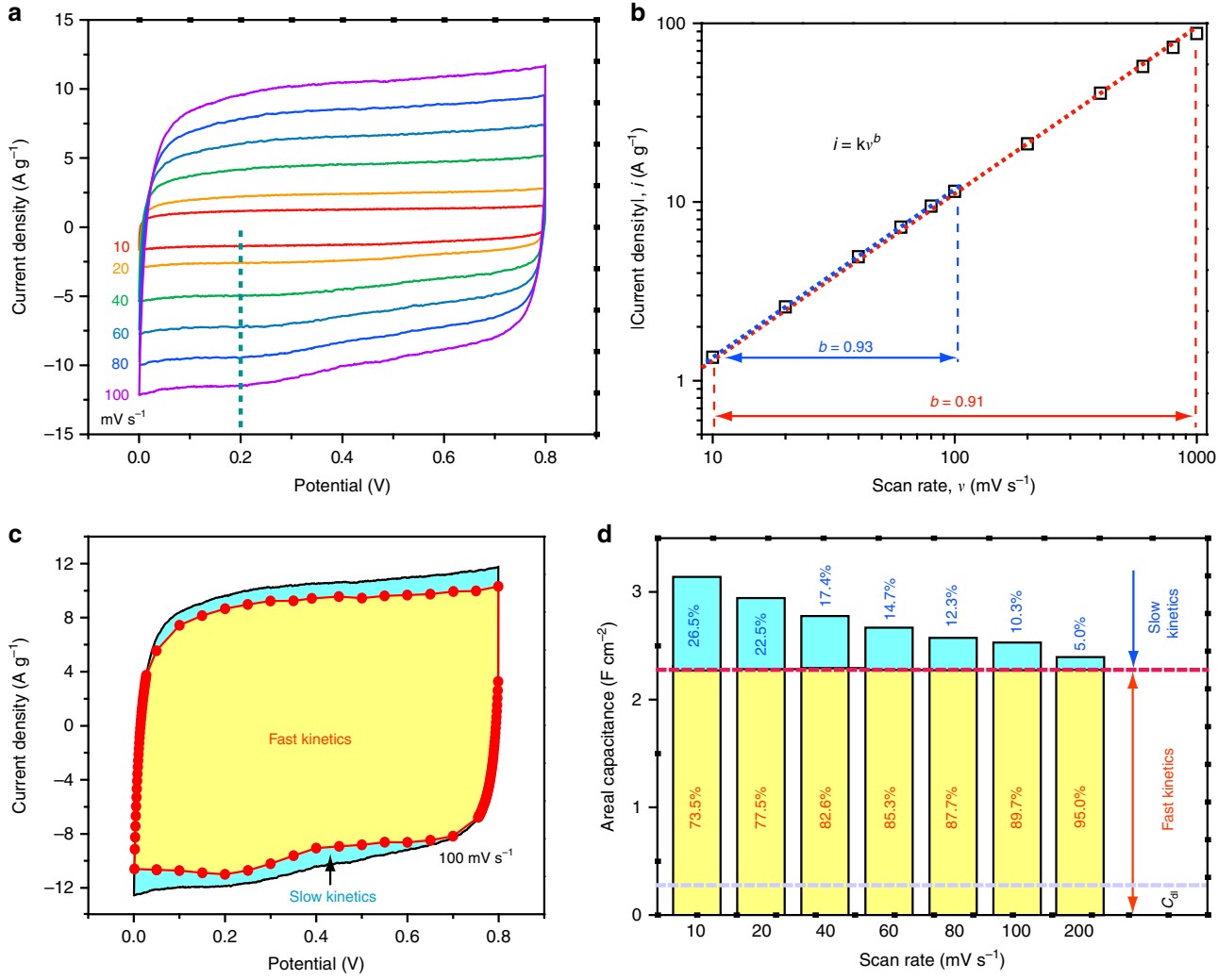

**Fig. 5** Ultra-fast charge-storage kinetics of PCF@MnO$_2$-2h. **a** CVs at various scan rates from 10 to 100 mV s$^{-1}$. The dashed line highlights the potential (0.2 V) selected for the $b$ value calculation. **b** Absolute current density and scan rate follow the power law, $i = kv^b$, in both the slow and fast scan rate regions. The dashed lines are best fitting lines and the $b$-value changes only slightly from the slow scan region to the fast scan region. **c** Decoupling of the capacitance contributed by the fast-kinetic processes (yellow) and the slow-kinetic processes (blue). Even at a high mass loading of MnO$_2$ (50% of the total mass), the fast-kinetics capacitance still dominates the overall capacitance. **d** Histograms of the capacitance contributions by the different processes: Yellow, fast-kinetic processes; blue, slow-kinetic processes; $C_{dl}$, electrical double layer capacitance

plotting the logarithm of the absolute cathodic current densities at 0.2 V against the logarithm of scan rates (Fig. 5b), the $b$-value was calculated to be 0.93 in the scan-rate range of 10-100 mV s$^{-1}$, approaching that of an ideal capacitor ($b = 1$) and suggesting the ultra-fast charge-storage kinetics. Outstandingly, the $b$ value decreases only slightly to 0.91 in the range of 10–1000 mV s$^{-1}$, unambiguously confirming its fast charge-storage kinetics.

We further decoupled the capacitances from fast-kinetic processes and slow-kinetic processes. The decoupling is based on the different contributions of fast and slow kinetics processes in the current density of a CV curve (see Supplementary Methods for details). Briefly, the current density at a fixed potential and a scan rate, $i$ is composed of two terms associated with the scan rate, $v$:

$$i = k_1 v + k_2 v^{0.5}, \qquad (1)$$

where $k_1$ and $k_2$ are constants. The first term $k_1 v$ equals the current density contributed from fast-kinetic processes and the second term $k_2 v^{0.5}$ is the current density associated with slow-

kinetic (or diffusion-controlled) processes. Dividing $v^{0.5}$ on both sides of Equation (1) gives:

$$i v^{-0.5} = k_1 v^{0.5} + k_2. \qquad (2)$$

Equation (2) shows that $iv^{0.5}$ and $v^{0.5}$ are expected to have a linear relationship, with $k_1$ and $k_2$ being the slope and the y-intercept, respectively. Repeating the above step at other scan rates reveals the current density contribution across the potential window and outlines the contribution from the fast-kinetic and slow-kinetic processes. Figure 5c shows an example of the decoupling of a CV at 100 mV s$^{-1}$. The capacitive contribution from the fast-kinetic processes (yellow region) clearly dominates that of the slow-kinetic processes (blue region) at all scan rates (Fig. 5c–d, Supplementary Fig. 10). The slow-kinetic capacitance decreased with the increasing scan rate. Importantly, the electric double layer capacitance ($C_{dl}$) contributed only a small fraction in the fast-kinetics region (Fig. 5d, gray dashed line), indicating that the majority of the pseudocapacitance of PCF@MnO$_2$-2h is not

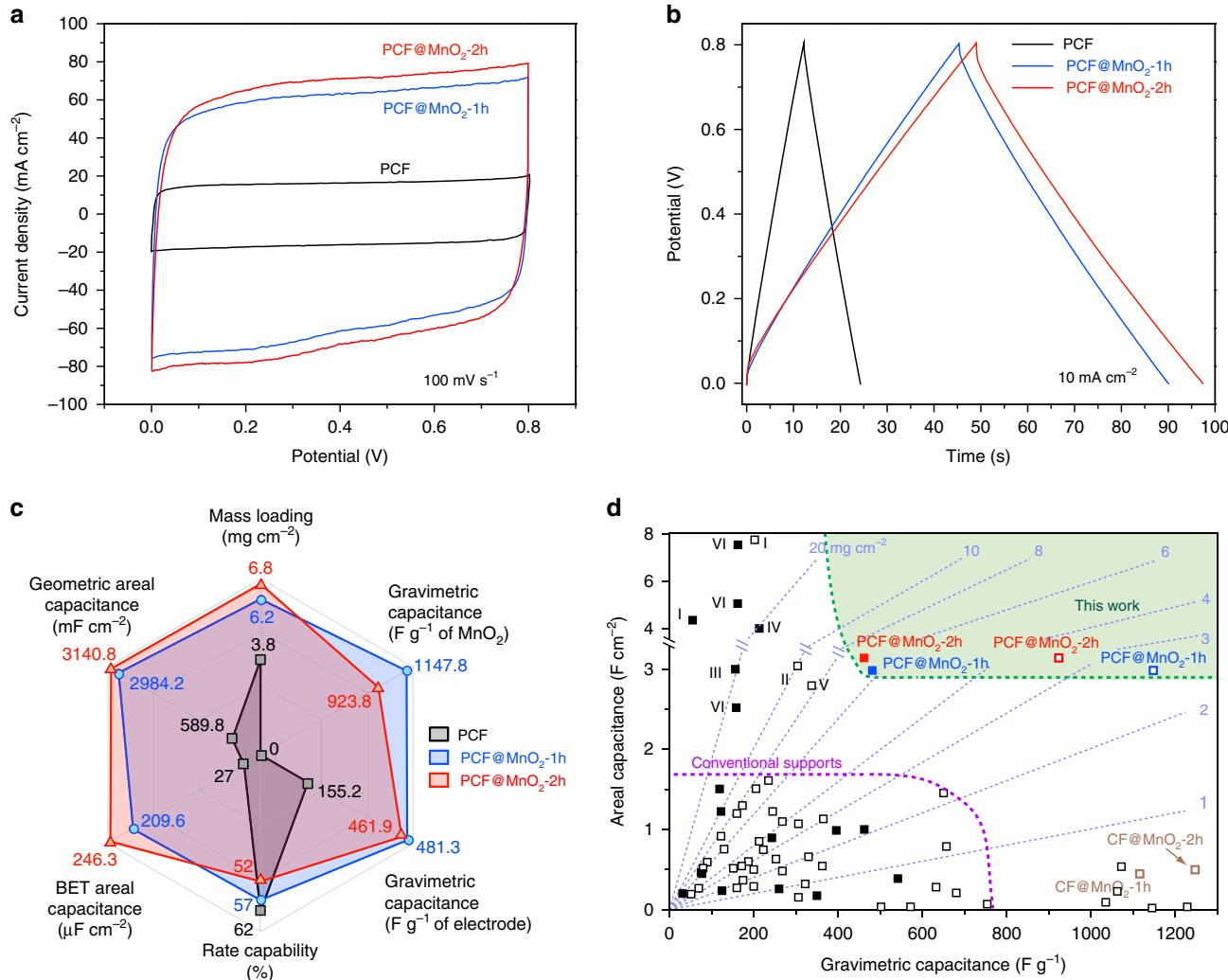

**Fig. 6** Electrochemical performance of PCF, PCF@MnO$_2$-1h and PCF@MnO$_2$-2h. **a** CVs at a scan rate of 100 mV s$^{-1}$. **b** Galvanostatic charge-discharge curves at 10 mA cm$^{-2}$ of PCF (black), PCF@MnO$_2$-1h (blue), and PCF@MnO$_2$-2h (red). **c** The radar chart compares the six figure-of-merits of PCF (black), PCF@MnO$_2$-1h (blue), and PCF@MnO$_2$-2h (red): mass loading of the active materials, rate capability (from 10 to 1000 mV s$^{-1}$), gravimetric capacitance based on the mass of MnO$_2$ and the active materials, and areal capacitance based on the geometric area and BET surface area. All capacitances are obtained at 10 mV s$^{-1}$. **d** Mass loading, gravimetric capacitance, and geometric areal capacitance of PCF-based electrodes in comparison with other reported electrodes. Dashed lines mark the mass loadings in mg cm$^{-2}$. Open and filled squares are capacitances based on the mass loadings of MnO$_2$ and the entire electrodes, respectively. Note that the open (and filled) squares are only to be compared with open (and filled) squares. The labeled points: I, wood-derived porous carbon@MnO$_2$;[14] II, hierarchical MnO$_2$ on carbon cloth;[12] III, carbon nanotube (CNT)@MnO$_2$;[60] IV, activated carbon-coated CNT@MnO$_2$;[61] V, CNT-wrapped polyester fiber@MnO$_2$;[16] VI, carbon nanofoam@MnO$_2$[33]. The details of all the unlabeled data points are summarized in Supplementary Table 2

charge-transfer-limited or diffusion-controlled. The fast kinetics makes PCF@MnO$_2$ a desirable pseudocapacitive electrode for rapid charge storage and release.

We measured the electrochemical capacitive performance of our carbon fiber electrodes. Among the PCF-based electrodes, PCF@MnO$_2$-2h displayed the highest areal capacitance, as shown by the CVs (Fig. 6a) and the galvanostatic charge-discharge (GCD) profiles (Fig. 6b). The negligible deviation of the CVs from the rectangular shape and the isosceles triangular GCD profiles at high 10 mA cm$^{-2}$ echoed the fast charge-storage kinetics of PCF@MnO$_2$. A radar chart (Fig. 6c) summarizes the six figure-of-merits of a pseudocapacitor electrode, *i.e.*, mass loading, gravimetric capacitance normalized to the mass of MnO$_2$, gravimetric capacitance normalized to the mass of electrode, areal capacitance normalized to geometric surface area, areal capacitance normalized to BET surface area, and rate capability. Due to the lower mass loading of PCF@MnO$_2$-1h than that of

PCF@MnO$_2$-2h, the former achieved higher values in gravimetric capacitance and rate capability. Remarkably, the gravimetric capacitance of PCF@MnO$_2$-1h at 10 mV s$^{-1}$ reached 1148 F g$^{-1}$ of MnO$_2$. This value is ~84% of the theoretical gravimetric capacitance of MnO$_2$ (1367 F g$^{-1}$) within a potential window of 0.8 V, even slightly higher than those on the model supports of mesoporous Au[22,23] and dendritic Ni[26], suggesting almost all the MnO$_2$ loaded on PCFs was accessible to the ions and contributed to the high capacitance. PCF@MnO$_2$-2h displayed the highest areal capacitance owing to its highest mass loading. PCF exhibited the best rate capability because it charges/discharges mostly via electrical double layers. Full comparison of the gravimetric, areal, and volumetric capacitances of PCF-based and CF-based electrodes at various scan rates are summarized in Supplementary Figs 11 and 12. Markedly, the gravimetric capacitance and geometric areal capacitance of PCF@MnO$_2$-2h outperformed the previously-reported MnO$_2$ electrodes at

comparable mass loadings under similar testing conditions (Fig. 6d and Supplementary Table 2). Ideally, with fast electron and ion transport at high mass loadings, both the areal and gravimetric capacitances are expected to be high. However, most reported $MnO_2$ electrodes have poor areal and/or gravimetric capacitances. In contrast, our PCF-supported $MnO_2$ electrodes have both high areal and gravimetric capacitances. PCF@$MnO_2$-2h was also highly stable, retaining more than 98% of the initial capacitance after 5000 consecutive charge-discharge cycles (Supplementary Fig. 13).

The Ragone plot (Supplementary Fig. 14) compares the specific energy and power densities of PCF@$MnO_2$ with those of the $MnO_2$ supported on graphene, a star material for supercapacitor electrodes. With a high gravimetric power density of 23.2 kW kg$^{-1}$ and a high gravimetric energy density of 10.3 Wh kg$^{-1}$ in the tested range of scan rates, PCF@$MnO_2$-2h outperformed the various graphene- and CF-supported $MnO_2$ electrodes in symmetric pseudocapacitors. The superior capacitive performance signifies that our PCF-supported $MnO_2$ electrodes have realized both high mass loadings and ultrafast charge transport kinetics.

## Discussion

The judiciously designed comparison between our PCFs and conventional CFs proves that PCFs with uniform mesopores are superior carbon supports for addressing the two long-lasting challenges of pseudocapacitors: high mass loading and fast charge transport. Utilizing the concept of block copolymer self-assembly and microphase separation, PCFs provide abundant mesopores with a large surface area for high mass loadings of ultrathin (<2 nm) pseudocapacitive materials. On the one hand, the ultrathin pseudocapacitive material, along with the continuous fibrous carbon network, renders the composite electrode fast electron transport. On the other hand, the partially filled mesopores provide continuous and wide-open channels for effective ion transport with little diffusion resistance, even at high mass loadings approaching 7 mg cm$^{-2}$. The PCF@$MnO_2$ electrodes show outstanding and balanced gravimetric capacitance, areal capacitance, and rate capability, which outperform other $MnO_2$-based pseudocapacitive electrodes at comparable mass loadings and testing conditions. Future investigations on the interplays among the polymer molecular weight, mesopore size, mass loading of $MnO_2$, ion diffusion resistivity and the use of ionic liquid electrolytes[55] are expected to further optimize the capacitive performance of PCF@$MnO_2$ and enhance the energy density of the supercapacitors.

This work signifies the great potential of leveraging the disparate and innovative concept of block copolymer microphase separation to design and fabricate mesoporous carbon fiber supports. We emphasize that the highly uniform mesopores are crucial for the high loading of guest materials and the efficient transport of ions. The block copolymer-derived PCFs revolutionize the porous carbon supports and are adaptable to a broad range of electrochemical applications including batteries, fuel cells, catalyst supports, and capacitive desalination devices.

## Methods

**Synthesis of porous carbon fiber mats**. Porous carbon fiber (PCF) mats were derived from poly(acrylonitrile-*block*-methylmethacrylate) (PAN-*b*-PMMA) block copolymer. Briefly, PAN-*b*-PMMA ($M_n = 110$-*b*-60 kDa, polydispersity = 1.14) was synthesized via reversible addition-fragmentation chain-transfer polymerization[56] and electrospun into a polymer fiber mat. The polymer fiber mat was cut into small stripes (e.g., 10 cm × 2 cm), loaded into a tube furnace (Thermo-Fisher Scientific, Model STF55433C-1), and then heated at 280 °C for 8 h (ramp rate: 1 °C min$^{-1}$) in air. The heating process induced the microphase separation of PAN and PMMA, and it triggered the crosslinking and cyclization of PAN. The resulting brown mats were further heated at 1200 °C for 1 h (ramp rate: 10 °C min$^{-1}$) under

a nitrogen atmosphere. Afterwards, the tube furnace was cooled down to room temperature and PCF mats were obtained. The preparation of CF was similar except that PAN was used instead of PAN-*b*-PMMA.

**Deposition of Manganese Dioxide**. Manganese dioxide ($MnO_2$) was deposited onto the PCF mats via a solution-based self-limiting redox reaction with potassium permanganate ($KMnO_4$),

$$4KMnO_4 + 3C + H_2O \rightarrow 4MnO_2 + K_2CO_3 + 2KHCO_3$$

First, 0.032 g of $KMnO_4$ powder was dissolved in 20 mL of deionized water and used as the deposition solution ($KMnO_4$, 10 mM). The solution was then heated to 80 °C under ambient pressure. Approximately 10 mg of PCF mats were soaked in the solution for 1–2 h under gentle stirring. After the reaction, the $KMnO_4$ solution was drained and the remaining carbon fiber mats were thoroughly washed with deionized water five times, followed by drying in a vacuum oven at 60 °C for 8 h. The resulting carbon fiber mats are designated as PCF@$MnO_2$-1h and PCF@$MnO_2$-2h based on the reaction times of 1 h and 2 h, respectively.

The mass loading of $MnO_2$ was determined by calculating the mass difference between the PCF mats before and after the reaction (see Supporting Information for detailed calculations). The areal mass loadings of $MnO_2$ in PCF@$MnO_2$-1h and PCF@$MnO_2$-2h were 2.6 ± 0.2 and 3.4 ± 0.4 mg cm$^{-2}$, respectively. The total mass loadings (including PCF and $MnO_2$) of PCF@$MnO_2$-1h and PCF@$MnO_2$-2h were 6.2 ± 0.3 and 6.8 ± 0.4 mg cm$^{-2}$, respectively. The average thickness of all PCF, PCF@$MnO_2$-1h and PCF@$MnO_2$-2h mats was ~200 μm. Thus, the volumetric mass densities of PCF@$MnO_2$-1h and PCF@$MnO_2$-2h were 0.31 ± 0.02 and 0.34 ± 0.02 g cm$^{-3}$, respectively. The standard deviations were based on at least three batches of carbon fiber based electrodes.

Electrochemical deposition was also adopted to prepare PCF@$MnO_2$ electrodes with high mass loadings. The electrodeposition solution contained 0.1 M manganese acetate and 0.5 M lithium chloride (a supporting electrolyte) in deionized water. A piece of PCF carbon fiber mat, a piece of nickel foam, and an Ag/AgCl wire in saturated KCl aqueous solution were used as the working electrode, the counter electrode, and the reference electrode, respectively. The electrodes were connected to an electrochemical workstation (PARSTATS 4000+, Princeton Applied Research, Ametek Inc.) and scanned between 0 and 1.0 V vs. Ag/AgCl at a scan rate of 0.01 mV s$^{-1}$ for 15 cycles. The mass loading of the electrodeposited $MnO_2$ on the PCF was 4.2 mg cm$^{-2}$. The total mass loading (including PCF and $MnO_2$) from electrodeposition was ~8.0 mg cm$^{-2}$.

**Physical Characterizations**. The carbon fibers were characterized using scanning electron microscopy (SEM, LEO Zeiss 1550, acceleration voltage: 2 kV) and high-resolution transmission electron microscopy (HRTEM, FEI TITAN 300, acceleration voltage: 300 kV). The physisorption isotherms were measured with a pore analyzer (3Flex Pore Analyzer, Micromeritics Instrument Corp.) using nitrogen (for mesopores) and carbon dioxide (for micropores). Prior to the sorption tests, all electrodes were heated at 90 °C for 60 min and then at 200 °C for 900 min in $N_2$ to desorb any moisture and hydrocarbons. The ramping rate of both heating processes was 10 °C min$^{-1}$. The surface areas were calculated using the Brunauer-Emmett-Teller (BET) method, and the pore size distributions were obtained by the density functional theory. X-ray photoelectron spectroscopy (XPS) spectra were acquired using monochromatic Al $K_\alpha$ X-ray source (1486.6 eV) with a 200 μm X-ray beam at an incident angle of 45°. All binding energies were referenced to adventitious C 1 s at 284.8 eV. Chemical states of elements were assigned based on the National Institute of Standards and Technology (NIST) XPS Database. Raman spectra were recorded by a Raman spectrometer (WITec alpha 500) coupled with a confocal Raman microscope using a laser excitation wavelength of 633 nm. UV-vis spectra were measured by an Agilent Cary 60 UV-vis spectrometer. Small angle X-ray scattering (SAXS) spectra were collected by a Bruker N8 Horizon instrument with Cu $K_\alpha$ radiation ($\lambda = 1.54$ Å) at a current of 1 mA and a generator voltage of 50 kV.

**Electrochemical characterizations**. The electrochemical performance was evaluated in a symmetric two-electrode configuration in an aqueous electrolyte of 6 M KOH. For consistency, carbon fiber mats were sandwiched between two pieces of nickel foams (EQ-bcnf-80 μm, MTI corporation). Cyclic voltammograms were collected within a potential window of 0–0.8 V at various scan rates of 10–1000 mV s$^{-1}$. Galvanostatic charge and discharge (GCD) were performed within the same potential window (0–0.8 V). Electrochemical impedance spectroscopy was conducted at open circuit potentials with frequencies between 0.1 Hz and 100 kHz with a perturbation of 5 mV. The CVs and EIS were recorded using a PARSTATS 4000 + electrochemical workstation (Princeton Applied Research, Ametek Inc.). The GCD curves were acquired from a charge-discharge cycler (Model 580, Scribner Associates Inc.).

## Data Availability

Data supporting the findings of this study are available in this paper and in the Supplementary Information or are available from the corresponding author upon reasonable request.

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

## Acknowledgements

This material is based upon work supported by the Air Force Office of Scientific Research under award number FA9550-17-1-0112 through the Young Investigator Program (YIP). Guoliang Liu acknowledges the American Chemical Society Petroleum Research Foundation for the Doctoral New Investigator (DNI) award. The authors acknowledge the use of electron microscopes in the Virginia Tech Institute for Critical Technology and Applied Science (ICTAS), and Dr. Xu Feng for assistance in the XPS analysis at the Surface Analysis Laboratory of Virginia Tech. The XPS is supported by the National Science Foundation under Grant No. CHE-1531834.

## Author contributions

G.L. and T.L. designed all the experiments. T.L. performed the electrochemical and physical tests. Z.Z. participated in the synthesis of block copolymers and helped with the electrochemical and physical sorption tests. Y.G. collected the SEM images. D.G. synthesized the PAN homopolymers (the CF precursors). G.L. and T.L. wrote the paper with input from all authors.

## Additional information

**Competing interests:** The authors declare no competing interests.

