## [Peer Review File · Nature Communications]

Reviewers' comments:

Reviewer #1 (Remarks to the Author):

This is a very interesting report demonstrating the uses of a well designed hierarchical support material to enable very high loading and near theoretical performance of MnO₂ in high rate supercapacitors. This work raises the bar for the performance of MnO₂ based materials. The manuscript is well written and the results are clearly described. I only have a couple minor comments / suggestions for the authors.

1. The authors attribute the improved performance to the uniform pore size. However, I would suspect that the connectivity between pores within the fibers is also important to maintain the high rate performance for ion transport. What morphology is developed for these materials?

2. The carbon source and carbonization temperature will dramatically impact the electrical conductivity (line 79 in proof). The authors should be careful with their comparison.

3. The authors are inconsistent in the pore size listed (~ 11 nm or ~ 11.7 nm). This is slightly confusing. Why is 11 nm the critical size? Can the authors offer a geometric argument for why this size might be important for the performance obtained?

4. line 124, the authors describe this as large scale production. What was the area/mass? What rate can these be produced (is this scalable)?

5. The authors use adverbs almost ad nauseam. I would suggest that the authors use these only where important points/feature are discussed to prevent these from being distracting. For example (line 162), the micropore volume experienced relatively small decrease (66%). I would state that 66% is a large reduction. The decrease in micropore volume is less than that of the mesopores, but this seems overstated. Similarly, line 236, 2 is drastically different from 7?

6. The charge transfer resistance of the -2h sample appears to be larger than the others, but the text states all were ~0.7 ohm. Is this correct? From eye, I would say that the -2h samples is ~ 1 ohm.

4.

Reviewer #2 (Remarks to the Author):

The manuscript reports the fabrication of MnO₂-loaded porous carbon fibers based on microphase segregation of block copolymers and their characteristics for pseudocapacitor applications. The authors claimed that the mesoporous structure of the porous carbon fibers allowed for high mass loading of MnO₂, fast charge/ion transport, high capacitances, which are beneficial characteristics for their applications in pseudocapacitors.

Overall, it is a well-written manuscript describing interesting results on a device application of block copolymers. Although the fabrication of mesoporous carbon has been previously reported by others using the same type of block copolymer, PAN-b-PMMA, I believe its utilization for MnO₂-based capacitors along with "excellent device performances" should present enough novelty. In general, the experiments were carefully done and the conclusions appear to be well-supported by the data. The manuscript may become publishable in Nature Comm after appropriate revisions.

1. Novelty: The fabrication of porous carbon fibers based on the carbonization of PAN-containing

block copolymers has been previously reported by others (Yan et al. *J. Mater. Chem. A.* 2015, 3, 22781; Song et al. *ACS Appl. Mater. Interface* 2018, 1, 2536). These previous articles somewhat reduce the novelty of the work, which makes me hesitant to recommend the publication of the work in *Nature Comm.* From the quick search, it doesn't seem like they were cited in the manuscript. Given the close relevance of the reports, they should be cited and properly introduced in the manuscript.

2. Is the block copolymer assembly structure drawn in Scheme 1 based on experimental data?

3. Is it possible to form different types of porous carbon structures using the block copolymers with different molecular weights or relative block lengths? How would it affect the device performances? I am not saying that such studies should be included in this manuscript, but I was curious if the polymer molecular weight etc have been optimized.

4. Figure 1 shows that MnO₂ continues to grow on PCF after 1h while the surface of CF@MnO₂-2h did not change significantly from CF@MnO₂-1h. I understand that PCF is likely to afford higher loading of MnO₂ due to the mesoporous structure. However, I do not see the connection between the internal mesostructure and the growth of MnO₂ on the outer surface of nanofibers.

5. The image quality of conventional CF (Figure 1d) is not as good as that of PCF (Figure 1a). I trust the authors in that the surface of CF is smoother than that of PCF. However, the smoothness of the CF surface might be somewhat exaggerated in the low quality image.

Reviewer #3 (Remarks to the Author):

Liu and co-workers have prepared multi-hierarchical porous carbon fibers with coated manganese dioxide (MnO₂). These materials were investigated as pseudocapacitor electrodes in a classical three electrode cell setup and in symmetric pseudocapacitor devices.

By fabricating high density mesopores embedded into carbon fibers with MnO₂ coatings, the authors have demonstrated electrode materials with large surface area values and low diffusivity resistance for ions. These results are substantiated by electrochemical impedance spectroscopy that analyze the low frequency impedance using a Warburg element. The high surface, excellent electrical conductivity, and low diffusion resistance of the multi-hierarchical carbon fibers translated to excellent capacitance, power density (gravimetric), and energy density values when compared against the best MnO₂ based supercapacitors supported on graphene/CNT substrates (note: when using aqueous based electrolytes).

The authors have taken a creative approach by making hierarchical porous structures by electrospinning poly(acrylonitrile-block-methyl methacrylate) block copolymers to yield 150 to 250 nm diameter fibers. The block copolymer fibers are then heated to induce microphase separation at the meso-length scale (i.e., periodic features of about 10 nm). Further heating and pyrolysis cause the PAN domain to crosslink and be electrically conductive. At the same time, the PMMA is thermally decomposed and removed to leave behind meso-porous fibers. The pores had diameters that ranged from 10 to 12 nm before MnO₂ was introduced. Electrodeposition of MnO₂ on the porous carbon fibers promoted ion diffusion and enabled pseudo-capacitance capability.

The implications of the work are far-reaching beyond supercapacitor electrodes. These high surface area, multi-hierarchical fibers will also be useful for fuel cell and battery applications and even ionic separations via capacitive deionization. Furthermore, the adoption of a block copolymer material enables the attainment of reliable structure-property relationships that correlate mesopore diameter to electrochemical properties (both reaction kinetic and diffusion limitations) of the electrode.

I strongly recommend for publication in Nature Communications after the authors address the following concerns:

1.) The morphology and period of the synthesized PAN-b-PMMA should be verified via SAXS.

2.) The authors have conclusively shown a uniform mesopore diameter from their BET N₂ adsorption isotherms, but do the fiber supports have perpendicular cylinder orientations of the block copolymer normal to the fiber strand or are the cylinders parallel to the fiber strand? If they are perpendicular, then the ions need to diffuse into the 10 nm pore. If they are parallel, then the ions do not need to diffuse as far into the pore. The greater length scale for diffusion will inevitably lead to larger mass transfer resistances.

Can the authors comment, or provide any statistical information, on the orientation of the cylinder forming block copolymers in the electrospun fibers?

3.) The discussion of the Nyquist plots in Figure 3 (line 198) should mention that the traces resemble mixed kinetic-diffusion controlled processes in the electrodes.

4.) From Figure 3a, why does the addition of MnO₂ increase the charge-transfer resistance (as evident by the diameter of the semi-circle)? I would suspect that more MnO₂ (2 hr versus 1 hr) would contain more active material and have a lower charge-transfer resistance.

5.) The presentation of Figure 4 and fast kinetics versus slow kinetics needs to be reworked. I suggest taking the equations in the SI (page 3) and moving them to the main manuscript. It wasn't clear in the main manuscript how Figure 4a was attained, but it is important to the overall claims and impact of the paper.

6.) If possible, I suggest the authors test their best pCNF-MnO₂ electrodes with ionic liquid electrolyte and to expand the voltage window during CV experiments. The selection of the aqueous 6 M KOH limited the power density and energy density values of the prepared pseudo-capacitance electrode materials. By removing the aqueous electrolyte, a voltage window can be expanded to up to 3 V. It would be nice to compare the performance properties of these materials against other promising capacitor devices featuring MnO₂ - as demonstrated by Zhang et al. - J. Mater. Chem. A. 2013 DOI: 10.1039/C3TA00981E

Reviewer #1:

This is a very interesting report demonstrating the uses of a well designed hierarchical support material to enable very high loading and near theoretical performance of MnO₂ in high rate supercapacitors (supercapacitors). This work raises the bar for the performance of MnO₂ based materials. The manuscript is well written and the results are clearly described. I only have a couple minor comments / suggestions for the authors.

Comment #1: *"The authors attribute the improved performance to the uniform pore size. However, I would suspect that the connectivity between pores within the fibers is also important to maintain the high rate performance for ion transport. What morphology is developed for these materials?"*

Response: We sincerely thank the reviewer for the kind words that our work "is a very interesting report" and "raises the bar for the performance of MnO₂ based materials". We also appreciate the reviewer's insightful comment on the connectivity of pores within the carbon fibers. We agree with the reviewer that the interconnected pores are important to maintain the high rate performance for ion transport. The volume fraction of PAN in PAN-*b*-PMMA was ~65% in this work. For a typical block copolymer, the morphology should be either cylindrical or gyroidal, depending on the incompatibility of the two blocks. However, because PAN was cross-linked during the thermal treatment at elevated temperatures, PAN-*b*-PMMA formed interconnected structures and the resultant pores were irregularly shaped, randomly oriented, and uniformly distributed in the carbon fibers, as illustrated in the cross-sectional SEM image (Figure S1a). We have attributed the irregular pore morphology to the crosslinking of PAN, which impedes the microphase separation of PAN-*b*-PMMA into well-defined cylinders or gyroids. The crosslinking-induced hindering effect on the block copolymer microphase separation is also reported in previous works (see examples, *Macromolecules*, 2006, 39, 4848-4859; *Macromolecules*, 2012, 45, 7590-7598).

Accordingly, we have revised the following sentences in the manuscript.

"Scanning electron microscopy (SEM) shows the contrasting morphologies of PCFs and CFs (Figures 1a, 1d and S1). Owing to the microphase separation of PAN-*b*-PMMA and the subsequent degradation of poly(methyl methacrylate) (PMMA), the PCFs were perforated with a large amount of uniformly distributed, randomly oriented, and interconnected mesopores of ~11.7 nm (Figures 1a, 2a, and S1a)⁴². In contrast, the CFs derived from PAN exhibited relatively smooth surfaces and no observable mesopores under SEM (Figures 1d and S1b)." [Page 6, revised main text]

"In addition, the σ value of PCF@MnO₂-2h ($< 2 \Omega \text{ s}^{-0.5}$) is ~3.5 times lower than that of CF@MnO₂-2h ($\sim 7 \Omega \text{ s}^{-0.5}$), highlighting the critical role of the uniform distributed, randomly

oriented, and interconnected mesopores in accelerating electrolyte infiltration and ion diffusion in block copolymer-derived PCFs." [Page 13, revised main text]

Figure S1 | (a) Representative cross-sectional SEM image of PCFs derived from PAN-*b*-PMMA. The PCFs show a large number of uniformly distributed, randomly oriented, and interconnected mesopores.

We have also added the following discussion to clarify the pore morphology of PCFs in the revised main text:

"The volume fraction of PAN in PAN-*b*-PMMA was ~65%, and supposedly the block copolymer should self-assemble into either cylindrical or gyroidal structures, depending on the incompatibility of the two blocks. After pyrolysis, however, the porous carbon fibers showed no well-defined cylindrical or gyroidal structures but interconnected mesopores that were irregularly shaped and uniformly distributed, as shown in the cross-sectional SEM image (Figure S1a). This morphology is attributed to the crosslinking of PAN at elevated temperatures, which hindered the microphase separation of PAN-*b*-PMMA into well-defined cylindrical or gyroidal structures, similar to the crosslinking-induced hindering effect in previous reports^{43,44}. " [Page 6, revised main text]

Comment #2: *"The carbon source and carbonization temperature will dramatically impact the electrical conductivity (line 79 in proof). The authors should be careful with their comparison."*

Response: We appreciate the reviewer's comment on the comparison of electrical conductivity. We agree with the reviewer that both the carbon source and processing conditions (*e.g.*, pyrolysis temperature) can change the electrical conductivities of the resultant carbons. Therefore, it is inappropriate to claim that the electrical conductivity of CMK is limited and we have removed the corresponding discussion. The revised text reads as follows:

"This contrasts with hard-templated mesoporous carbon particulates (e.g., CMK-3³¹), which ~~have limited electrical conductivity^{5,15}~~ and generally demand polymer binders to hold the discrete carbon particulates together." [Page 5, revised main text]

Comment #3: "The authors are inconsistent in the pore size listed (~ 11 nm or ~ 11.7 nm). This is slightly confusing. Why is 11 nm the critical size? Can the authors offer a geometric argument for why this size might be important for the performance obtained?"

Response: We are sorry for the pore size inconsistency. According to the N₂-physisorption test, the mesopore size of the block copolymer-derived PCFs have a peak at 11.7 nm. For consistency, we have used the precise value of 11.7 nm throughout our manuscript. Revisions include:

"Utilizing block copolymers that microphase-separate, we synthesize porous carbon fibers (PCFs) with uniform mesopores of 11.7 nm, which are partially filled with MnO₂ of <2 nm in thickness." [Page 2, revised main text]

"Importantly, our judiciously designed mesopores have an average diameter of 11.7 nm and are partially filled with a <2-nm-thick layer of MnO₂ (Scheme 1c)." [Page 4, revised main text]

In terms of the performance, we do not think that the precise value of 11.7 nm is critical, and any pore size around 11.7 nm should lead to a similar performance. However, the following conditions must be met: The pores must remain open and accessible to ions after depositing MnO₂, and the remaining pores must be effective to facilitate ion diffusion. In this work, we have not optimized the mesopore size of PCF yet, and we plan to study the interplay among pore size, mass loading and ion diffusion resistivity, which will be the subject of a full paper in the future.

Accordingly, we have added the following discussion in the revised main text:

"The comparison shows that the uniform mesopores are indispensable in realizing the high mass loading of MnO₂ on the carbon fibers. The abundant mesopores provide large solution-accessible surface areas for loading MnO₂ on the PCFs, while micropores can only host a limited amount of MnO₂ because the deposition solution can barely access them." [Page 12, revised main text]

"In addition, the σ value of PCF@MnO₂-2h (< 2 Ω s^{-0.5}) is ~3.5 times lower than that of CF@MnO₂-2h (~7 Ω s^{-0.5}), highlighting the critical role of the uniform distributed, randomly oriented, and interconnected mesopores in accelerating electrolyte infiltration and ion diffusion in block copolymer-derived PCFs. The relatively large pore openings make the mesopores less vulnerable to MnO₂ clogging than micropores, ensure the ion/electrolyte diffusion channels unobstructed, and maintain a low ion diffusion resistance." [Page 14, revised main text]

"The PCF@MnO₂ electrodes show outstanding and balanced gravimetric capacitance, areal capacitance, and rate capability, which outperform other MnO₂-based pseudocapacitive electrodes at comparable mass loadings and testing conditions. Future investigations on the interplays among the polymer molecular weight, mesopore size, mass loading of MnO₂, and ion

diffusion resistivity are expected to further optimize the capacitive performance of PCF@MnO₂." [Page 19, revised main text]

Comment #4: "line 124, the authors describe this as large scale production. What was the area/mass? What rate can these be produced (is this scalable)?"

Response: We appreciate the reviewer's comment on the production scale. In one polymerization batch reactor, we typically synthesize ~10 g of PAN-*b*-PMMA block copolymer. Each electrospinning process consumes about 1 g of polymers and produces a polymer fiber mat with an area of ~45 cm×15 cm (Figure R1). The subsequent pyrolysis of PAN-*b*-PMMA fiber mat into porous carbon fibers follows the standard industrial practice of carbon fiber preparation. After pyrolysis, the deposition of MnO₂ on carbon fibers to obtain PCF@MnO₂-2h takes ~2 hours.

In our laboratory, the production rate is mainly restricted by the size of electrospinning mats and the discontinuous batch-by-batch pyrolysis processing. We expect to significantly further enhance the production rate and capacity by adopting roll-to-roll fiber preparation and pyrolysis. We have identified facilities at Oak Ridge National Laboratory (ORNL) and industrial partners. We plan to implement the large-scale production of such materials in the near future.

Figure R1 | A photograph of a PAN-*b*-PMMA fiber (~45×15 cm²) mat electrospun on a piece of Al foil.

Comment #5: "The authors use adverbs almost ad nauseam. I would suggest that the authors use these only where important points/feature are discussed to prevent these from being distracting. For example (line 162), the micropore volume experienced relatively small decrease (66%). I would state that 66% is a large reduction. The decrease in micropore volume is less than that of the mesopores, but this seems overstated. Similarly, line 236, 2 is drastically different from 7?"

Response: We appreciate the reviewer's comment on the use of adverbs. Following the suggestion, we have thoroughly checked the manuscript and made the following revisions (the revised parts are underlined or crossed out):

"The pore size distributions of PCFs and CFs, however, were ~~drastically~~ different in the mesopore range." [Page 9, revised main text]

"As shown in Table S1, the pore volume reduced more in the mesopore range (86.1% reduction after 2-h deposition) than in the micropore range (66.0% reduction after the 2-h deposition)." [Page 9-10, revised main text]

"The combined series resistances (R_s) of PCF and PCF@MnO₂-1h were ~~as small as~~ 1.0 Ω , and that of PCF@MnO₂-2h increased ~~slightly~~ to 1.4 Ω (Figure 3a inset)."

"The ~~significantly~~ higher loadings of MnO₂ in PCFs than in CFs is due to the large amount of uniform mesopores (Figure 2d)." [Page 12, revised main text]

"In addition, the σ value of PCF@MnO₂-2h ($< 2 \Omega \text{ s}^{-0.5}$) is ~ 3.5 times lower than that of CF@MnO₂-2h ($\sim 7 \Omega \text{ s}^{-0.5}$), highlighting the critical role of the uniform distributed, randomly oriented, and interconnected mesopores in accelerating electrolyte infiltration and ion diffusion in block copolymer-derived PCFs." [Page 14, revised main text]

"Due to the ~~slightly~~ lower mass loading of PCF@MnO₂-1h than that of PCF@MnO₂-2h, the former achieved higher values in gravimetric capacitance and rate capability." [Page 16, revised main text]

Comment #6: "The charge transfer resistance of the -2h sample appears to be larger than the others, but the text states all were ~ 0.7 ohm. Is this correct? From eye, I would say that the -2h samples is ~ 1 ohm."

Response: We have double-checked the EIS fitting results. The charge-transfer resistances of PCF, PCF@MnO₂-1h and PCF@MnO₂-2h are 0.74, 0.86 and 1.30 Ω , respectively. We have revised the discussion as follows.

"In addition, the charge-transfer resistances (R_{ct} , the semicircles in Figure 3a inset) of PCF, PCF@MnO₂-1h and PCF@MnO₂-2h are 0.74, 0.86 and 1.30 Ω , respectively. The small resistances suggest efficient electron transfer associated with the redox reaction of MnO₂. The augmentation of charge-transfer resistance in PCF@MnO₂-2h is mainly due to the increased thickness of MnO₂ deposited in the mesopores (as evidenced by the reduction in mesopore-width, Figure 2d). The increased thickness elongates the electron transport distance in MnO₂ and therefore obstructs electron transfer at the MnO₂/electrolyte interface, because MnO₂ is a poor electron conductor ($10^{-5}\sim 10^{-6} \text{ S cm}^{-1}$)." [Page 13, revised main text]

Reviewer #2:

The manuscript reports the fabrication of MnO₂-loaded porous carbon fibers based on microphase segregation of block copolymers and their characteristics for pseudocapacitor applications. The authors claimed that the mesoporous structure of the porous carbon fibers allowed for high mass loading of MnO₂, fast charge/ion transport, high capacitances, which are beneficial characteristics for their applications in pseudocapacitors.

*Overall, it is a well-written manuscript describing interesting results on a device application of block copolymers. Although the fabrication of mesoporous carbon has been previously reported by others using the same type of block copolymer, PAN-*b*-PMMA, I believe its utilization for MnO₂-based capacitors along with "excellent device performances" should present enough novelty. In general, the experiments were carefully done and the conclusions appear to be well-supported by the data. The manuscript may become publishable in Nature Comm after appropriate revisions.*

Comment #1: *"Novelty: The fabrication of porous carbon fibers based on the carbonization of PAN-containing block copolymers has been previously reported by others (Yan et al. J. Mater. Chem. A. 2015, 3, 22781; Song et al. ACS Appl. Mater. Interface 2018, 1, 2536). These previous articles somewhat reduce the novelty of the work, which makes me hesitant to recommend the publication of the work in Nature Comm. From the quick search, it doesn't seem like they were cited in the manuscript. Given the close relevance of the reports, they should be cited and properly introduced in the manuscript."*

Response: We thank the reviewer for bringing the two papers to our attention. We find the first paper by Yan *et al.* but are unable to acquire the second paper by Song *et al.* in *ACS Appl. Mater. Interface* in the literature. After diligent research, we suspect the most possible paper that the reviewer may refer to is "Song *et al.* *ACS Appl. Nano Mater.* **2018**, 1, 2536-2543". If this is not the correct article, we welcome the reviewer to provide additional information to help us identify it.

Although both works may involve block copolymer-derived carbons, our work distinguishes from the work by Yan *et al.* and the work by Song *et al.* First, the porous carbon fiber is an important form factor that contributes to the performance of our mesoporous carbons, while Yan *et al.* have synthesized porous carbon in the form of powders. The self-supporting and continuously conductive nature of porous carbon fibers eliminates the needs for polymer binders and conductive additives for electrode preparation. In contrast, Yan *et al.* used 5 wt% polytetrafluoroethylene as a binder, as well as 7.5 wt% graphite and 7.5 wt% acetylene black as conductive additives. The incorporation of these components is undesirable because they increase the inert mass of electrodes and lower the gravimetric capacitance of the devices. Additionally, the additives may clog the pores and reduce the accessibility of the ions to the pores. Second, the focus of this communication is to highlight the potential of block copolymer-derived porous carbon fibers in promoting the charge-storage performance of pseudocapacitors

with high mass loadings of pseudocapacitive materials. The scope of the work by Yan *et al.*, however, is mainly about carbon electrodes for electrical double layer capacitors. The work by Song *et al.* in *ACS Appl. Nano Mater.* very nicely demonstrates the preparation of porous carbon powders from polyacrylonitrile-*block*-poly(*n*-butyl acrylate), which are utilized as sorbents for capturing Cr⁶⁺ and U⁶⁺ heavy metal ions. Both the form factor of the carbon materials and the targeted application are different from our porous carbon fibers and their use as supports for pseudocapacitive MnO₂. We therefore believe that the existing literature do not reduce the novelty of our work.

Nevertheless, we thank the reviewer for kindly pointing out these excellent papers. We have cited the paper by Song *et al.* and Yan *et al.* as references #40 and #41, respectively, in the Introduction section to highlight the block copolymer's capability of generating porous carbon materials. The two papers are cited in the following sentence:

"Because block copolymers self-assemble and microphase separate into uniform and continuous nanoscale domains³⁴⁻⁴¹, after pyrolysis they generate interconnected mesoporous carbons with large surface areas for depositing MnO₂." [Page 4, revised main text]

Comment #2: "Is the block copolymer assembly structure drawn in Scheme 1 based on experimental data?"

Response: We appreciate the reviewer's comment on Scheme 1. For better illustration of the structures in the PCFs, we have updated the Scheme. The new scheme highlights the mesopores on fiber surfaces observed by SEM, and the interconnected pores inside the fibers that are quantitatively verified in our previous work [DOI: 10.1126/sciadv.aau6852]. The updated scheme illustrates more accurately the actual morphologies in our porous carbon fibers.

Scheme | Schematic illustration of the synthesis of PCF and PCF@MnO₂. (a) PAN-*b*-PMMA block copolymer is spun into a polymer fiber mat. (Magnified view) PAN (blue) and PMMA (red) in the block copolymer fibers microphase-separate into a disordered bicontinuous network structure. Via thermal oxidation, PAN is crosslinked to ensure a high yield of conductive carbon network. (b) Upon pyrolysis, the polymer fibers are converted to porous

carbon fibers (PCFs, black) with continuous and uniform mesopores (white channels), which afford high loadings of transition metal oxides. (c) The PCFs are loaded with MnO₂ (magenta) to become PCF@MnO₂ through a solution-based redox deposition reaction. (Magnified view) The continuous carbon fiber matrix and the partially filled mesoporous channels provide effective “expressways” for electron conduction and ion diffusion, respectively.

Comment #3: *"Is it possible to form different types of porous carbon structures using the block copolymers with different molecular weights or relative block lengths? How would it affect the device performances? I am not saying that such studies should be included in this manuscript, but I was curious if the polymer molecular weight etc. have been optimized."*

Response: We appreciate the reviewer's questions on the interplays among the molecular weight or block lengths, the pore structure, and the device performance. The molecular weight of PAN-*b*-PMMA and the pore morphology have not been optimized in this work. The investigation of the relationship between the molecular weight and porous structure of PCF is ongoing and will be presented in a forthcoming full article. Our preliminary results to date show that an increase in the volume fraction of PMMA from 25% to ~75% expands the mesopore size from ~10 to 20 nm. An increase in the molecular weight will also change the pore sizes and thus performance. In short, more details will be reported soon and we have not optimized the molecular weight in this communication.

Comment #4: *"Figure 1 shows that MnO₂ continues to grow on PCF after 1h while the surface of CF@MnO₂-2h did not change significantly from CF@MnO₂-1h. I understand that PCF is likely to afford higher loading of MnO₂ due to the mesoporous structure. However, I do not see the connection between the internal mesostructure and the growth of MnO₂ on the outer surface of nanofibers."*

Response: We appreciate the reviewer's insightful comment on the growth of MnO₂. Based on the work by Chodankar *et al.* [*RSC Adv.*, 2014, 4, 61503-61513], we hypothesize that the coalescence of MnO₂ accounts for the larger and thicker MnO₂ nanosheets of PCF@MnO₂-2h than those of CF@MnO₂-2h. On PCF, the vertically grown MnO₂ nanosheets [*ACS Nano*, 2018, 12, 1033-1042] tend to orient randomly towards one another due to the curved pore surfaces. The short distances among the MnO₂ nanosheets (red circles in Figure R2a) promotes the coalescence among adjacent MnO₂ sheets. On the contrary, because CFs have relatively smooth surfaces, the MnO₂ nanosheets on CFs are unlikely to lean towards one another and hence the coalescence process is not as significant, resulting thinner and small MnO₂ grains (Figure R2b).

The above hypothesis is similar to the growth of graphene sheets on carbon fibers with rough surfaces [*Adv. Mater.*, 2018, 30, 170538]. Via chemical vapor deposition, large flakes of graphene nanosheets grow on surface-etched porous carbon fibers and form a morphology similar to PCF@MnO₂-2h (Figure R3a). On un-etched fibers with smooth surfaces, however, the same deposition method only yields graphene particles or overcoatings (Figure R3b).

Figure R2 | Schematic illustrations of MnO₂ nanosheets deposited on (a) PCF and (b) CF. The circles in panel a highlight the possible positions where coalescence occurs.

Figure R3 (reproduced from *Adv. Mater.*, 2018, 30, 170538) | The morphologies of graphene grown on (a) an etched carbon fiber and (b) a smooth carbon fiber via chemical vapor deposition.

Comment #5: "The image quality of conventional CF (Figure 1d) is not as good as that of PCF (Figure 1a). I trust the authors in that the surface of CF is smoother than that of PCF. However, the smoothness of the CF surface might be somewhat exaggerated in the low quality image."

Response: We appreciate the reviewer's comment on the quality of the SEM image in Figure 1d. We have re-collected the SEM images of CFs (Figure R1) and replaced the original ones. The new SEM images show that the surface of CFs is relatively smooth and devoid of mesopores. We have also revised the discussion on CF surface morphology as follows:

"In contrast, the CFs derived from PAN exhibited relatively smooth surfaces and no observable mesopores under SEM (Figures 1d and S1b)." [Page 6, revised main text]

Figure 1d | SEM images of (d) conventional CF. (Inset) A magnified view of a single CF fiber surface.

Reviewer #3:

Liu and co-workers have prepared multi-hierarchical porous carbon fibers with coated manganese dioxide (MnO₂). These materials were investigated as pseudocapacitor electrodes in a classical three electrode cell setup and in symmetric pseudocapacitor devices.

By fabricating high density mesopores embedded into carbon fibers with MnO₂ coatings, the authors have demonstrated electrode materials with large surface area values and low diffusivity resistance for ions. These results are substantiated by electrochemical impedance spectroscopy that analyze the low frequency impedance using a Warburg element. The high surface, excellent electrical conductivity, and low diffusion resistance of the multi-hierarchical carbon fibers translated to excellent capacitance, power density (gravimetric), and energy density values when compared against the best MnO₂ based supercapacitors supported on graphene/CNT substrates (note: when using aqueous based electrolytes).

The authors have taken a creative approach by making hierarchical porous structures by electrospinning poly(acrylonitrile-block-methyl methacrylate) block copolymers to yield 150 to 250 nm diameter fibers. The block copolymer fibers are then heated to induce microphase separation at the meso-length scale (i.e., periodic features of about 10 nm). Further heating and pyrolysis cause the PAN domain to crosslink and be electrically conductive. At the same time, the PMMA is thermally decomposed and removed to leave behind meso-porous fibers. The pores had diameters that ranged from 10 to 12 nm before MnO₂ was introduced. Electrodeposition of MnO₂ on the porous carbon fibers promoted ion diffusion and enabled pseudo-capacitance capability.

The implications of the work are far-reaching beyond supercapacitor electrodes. These high surface area, multi-hierarchical fibers will also be useful for fuel cell and battery applications and even ionic separations via capacitive deionization. Furthermore, the adoption of a block copolymer material enables the attainment of reliable structure-property relationships that correlate mesopore diameter to electrochemical properties (both reaction kinetic and diffusion limitations) of the electrode.

I strongly recommend for publication in Nature Communications after the authors address the following concerns:

Comment #1: *The morphology and period of the synthesized PAN-b-PMMA should be verified via SAXS.*

Response: We appreciate the reviewer's suggestion on SAXS characterization. Following this advice, we have collected SAXS spectra and the associated SEM images to reveal the pore morphologies of the as-spun PAN-*b*-PMMA, oxidized PAN-*b*-PMMA and PCF. Under SEM, the surface of the as-spun PAN-*b*-PMMA fiber is wrinkled but devoid of any distinct features (Figure S2a), while those of the oxidized PAN-*b*-PMMA and PCF contain dark domains (Figures S2, b and c). Because PMMA in PAN-*b*-PMMA degrades at ~320 °C-430 °C (Small, 2017, 13, 1603107), the bright and dark domains in oxidized PAN-*b*-PMMA are PAN and PMMA,

respectively. Correspondingly, the bright and dark regions in PCFs are PAN-derived carbon and pores, respectively.

The SAXS spectra corroborate the SEM images. The SAXS spectrum of PAN-*b*-PMMA is featureless, confirming that PAN-*b*-PMMA lacks well-defined microstructures. After oxidation, a broad Bragg peak appears at $\sim 0.196 \text{ nm}^{-1}$, corresponding to an average center-to-center pore-spacing of 32.0 nm. After pyrolysis at 1200 °C, the peak shifts to $\sim 0.244 \text{ nm}^{-1}$ and the average center-to-center pore-spacing reduces to 25.7 nm. The reduction in the pore-spacing is resulted from the shrink of the materials after pyrolysis. The broad SAXS peaks are due to the cross-linking of PAN, which prevents PAN-*b*-PMMA from forming well-defined cylindrical or gyroidal structures (see more details in the Response to Comment #1 of Reviewer #1).

Accordingly, we have added the following discussion and Figure S2 in the revised Supplementary Information.

“Under SEM, the surfaces of the as-spun PAN-*b*-PMMA fibers are wrinkled but devoid of any distinct feature (Figure S2a), while those of the oxidized PAN-*b*-PMMA and PCFs contain dark domains (Figures S2, b and c). Because PMMA in PAN-*b*-PMMA degrades at $\sim 320 \text{ °C}$ - 430 °C (ref.³), the bright and dark domains in the oxidized PAN-*b*-PMMA are PAN and PMMA, respectively. Correspondingly, the bright and dark regions in PCFs are PAN-derived carbon and pores, respectively.

The SAXS spectra corroborate the SEM images. The SAXS spectrum of PAN-*b*-PMMA is featureless, confirming that PAN-*b*-PMMA lacks well-defined microstructures. After oxidation, a broad Bragg peak appears at $\sim 0.196 \text{ nm}^{-1}$, corresponding to an average center-to-center pore-spacing of 32.0 nm. After pyrolysis at 1200 °C, the peak shifts to $\sim 0.244 \text{ nm}^{-1}$ and the average center-to-center pore-spacing reduces to 25.7 nm. The reduction in the pore-spacing is resulted from the shrink of the materials after pyrolysis. The broad SAXS peaks are due to the cross-linking of PAN, which prevents PAN-*b*-PMMA from forming well-defined cylindrical or gyroidal nanostructures.”

The following discussion on the SAXS results is incorporated in the revised main text.

"Small angle X-ray scattering (SAXS) spectroscopy confirmed the microphase separation of PAN-*b*-PMMA and revealed that the average center-to-center pore-spacing in PCFs was 25.7 nm (Figure S2)." [Page 6, revised main text]

The following technical details of SAXS is included in the "Physical Characterizations" section of the revised main text.

Small angle X-ray scattering (SAXS) spectra were collected by a Bruker N8 Horizon instrument with Cu $K\alpha$ radiation ($\lambda=1.54 \text{ \AA}$) at a current of 1 mA and a generator voltage of 50 kV. [Page 20, revised main text]

The following equation of the center-to-center pore spacing is added in the "Calculations" section of the revised Supplementary Information.

The center-to-center pore-spacing, d (in nm), was estimated based on the SAXS spectra:

$$d = \frac{2\pi}{q}$$

where q is the characteristic scattering vector (nm^{-1}).

Figure S2 | Morphologies and SAXS spectra. (a-c) SEM images of (a) as-spun PAN-*b*-PMMA, (b) oxidized PAN-*b*-PMMA and (c) PCF fibers. (d) SAXS spectra of as-spun PAN-*b*-PMMA, oxidized PAN-*b*-PMMA, and PCF fibers.

Comment #2: *The authors have conclusively shown a uniform mesopore diameter from their BET N_2 adsorption isotherms, but do the fiber supports have perpendicular cylinder orientations of the block copolymer normal to the fiber strand or are the cylinders parallel to the fiber strand? If they are perpendicular, then the ions need to diffuse into the 10 nm pore. If they are parallel, then the ions do not need to diffuse as far into the pore. The greater length scale for diffusion will inevitably lead to larger mass transfer resistances.*

Can the authors comment, or provide any statistical information, on the orientation of the cylinder forming block copolymers in the electrospun fibers?

Response: We sincerely thank the reviewer's question about the orientation of the cylindrical pores in the carbon fibers. The volume fraction of PAN in the PAN-*b*-PMMA used here is ~65%. For a typical block copolymer, the self-assembled morphology is expected to be either

cylindrical or gyroidal, depending on the incompatibility between the two blocks. However, the block copolymer used in this work is PAN-*b*-PMMA and it differs from typical block copolymers. Since PAN was cross-linkable during the thermal treatment at elevated temperatures, PAN-*b*-PMMA was unable to form well-defined cylindrical or gyroidal structures. Instead, it formed a mesoporous structure with interconnected and randomly distributed pores in the carbon fibers, as illustrated in the cross-sectional SEM image (Figure S1a). Therefore, the orientation of the pores is random and there is no preference to either parallel or perpendicular orientation to the fiber strands. As shown in the SEM image (Figure 1), the pores have a large array of openings on the carbon fiber skin surfaces. These open and interconnected pores allow for facile mass transfer into and inside the carbon fibers.

Figure S1. (a) Representative cross-sectional SEM images of PAN-*b*-PMMA-derived PCFs. The PCFs show a large number of interconnected and randomly oriented mesopores.

We have added the following to clarify the pore orientation in the revised main text:

"The volume fraction of PAN in PAN-*b*-PMMA was ~65%, and supposedly the block copolymer should self-assemble into either cylindrical or gyroidal structures, depending on the incompatibility of the two blocks. After pyrolysis, however, the porous carbon fibers showed no well-defined cylindrical or gyroidal structures but interconnected mesopores that were irregularly shaped and uniformly distributed, as shown in the cross-sectional SEM image (Figure S1a). This morphology is attributed to the crosslinking of PAN at elevated temperatures, which hindered the microphase separation of PAN-*b*-PMMA into well-defined cylindrical or gyroidal structures, similar to the crosslinking-induced hindering effect in previous reports^{43,44}. " [Page 6, revised main text]" [Page 6, revised main text]

Comment #3: *The discussion of the Nyquist plots in Figure 3 (line 198) should mention that the traces resemble mixed kinetic-diffusion controlled processes in the electrodes.*

Response: We gratefully thank the reviewer for the kind suggestion on discussing the Nyquist plots. We have revised the discussion and added the suggested information (underlined).

"The Nyquist plots of PCF, PCF@MnO₂-1h and PCF@MnO₂-2h (Figure 3a) exhibited incomplete semicircles followed by linear tails, which resemble the features of mixed kinetic-diffusion controlled processes and are typical for pseudocapacitive materials⁵⁰. " [Page 12, revised main text]

Comment #4: *From Figure 3a, why does the addition of MnO₂ increase the charge-transfer resistance (as evident by the diameter of the semi-circle)? I would suspect that more MnO₂ (2 hr versus 1 hr) would contain more active material and have a lower charge-transfer resistance.*

Response: We appreciate the reviewer's question about the charge-transfer resistance. We think that the slight augmentation of charge-transfer resistance in PCF@MnO₂-2h is mainly due to the increased thickness of MnO₂ deposited in the mesopores, as evidenced by the reduction in mesopore-width shown in Figure 2d. The reasons are as follows. The thick layer of MnO₂ elongates the electron transport distance and therefore obstructs electron transfer at the MnO₂/electrolyte interface. Because MnO₂ is a poor electron conductor ($10^{-5}\sim 10^{-6}$ S cm⁻¹), the "apparent resistance" increases and the charges are not as easily transferred. In addition, as the thickness increases, the surface to volume ratio decreases, which further reduces the opportunity for charge transfer. This phenomenon of increasing charge-transfer resistance is similar to those reported in previous works, where charge-transfer resistance also increased with the thickness or the mass loading of MnO₂ (*ACS Nano*, 2018, 12, 3557-4567; *Int. J. Electrochem. Sci.*, 2014, 9, 4024-4038; *Electrochim. Acta*, 2012, 78, 515-523). Detailed mechanism for this phenomenon may need additional comprehensive studies and can be a subject of future work.

Accordingly, the following discussion is added in the main text.

"...In addition, the charge transfer resistances (R_{ct} , the semicircles in Figure 3a inset) of PCF, PCF@MnO₂-1h and PCF@MnO₂-2h are 0.74, 0.86 and 1.30 Ω , respectively. These small resistances suggest efficient electron transfer associated with the redox reaction of MnO₂. The slight augmentation of charge-transfer resistance in PCF@MnO₂-2h is mainly due to the increased thickness of MnO₂ deposited in the mesopores (as evidenced by the reduction in mesopore-width, Figure 2d). The increased thickness elongates the electron transport distance in MnO₂ and therefore, obstructs electron transfer at the MnO₂/electrolyte interface because MnO₂ is a poor electron conductor ($10^{-5}\sim 10^{-6}$ S cm⁻¹)." [Page 13, revised main text]

Comment #5: *The presentation of Figure 4 and fast kinetics versus slow kinetics needs to be reworked. I suggest taking the equations in the SI (page 3) and moving them to the main manuscript. It wasn't clear in the main manuscript how Figure 4a was attained, but it is important to the overall claims and impact of the paper.*

Response: We appreciate the reviewer's insightful and constructive comment on the presentation of Figure 4. Because Figure 4a depicts a series of cyclic voltammograms, it is difficult to

distinguish the fast-kinetic and slow-kinetic contributions for all of them. We think that the reviewer may refer to Figure 4c and we take the liberty of discussing Figure 4c instead. Following the reviewer's suggestion, we have re-organized the discussion about Figure 4c and moved the equations in the SI (page 3) into the main manuscript, as follows:

"We further decoupled the capacitances from fast-kinetic processes and slow-kinetic processes. The decoupling is based on the different contributions of fast and slow kinetics processes in the current density of a CV curve (see the "Capacitance Differentiation" section in Supplementary Information for details). Briefly, the current density at a fixed potential and a scan rate, i is composed of two terms associated with the scan rate, v :

$$i = k_1 v + k_2 v^{0.5} \quad (1)$$

where k_1 and k_2 are constants. The first term $k_1 v$ equals the current density contributed from fast-kinetic processes and the second term $k_2 v^{0.5}$ is the current density associated with slow-kinetic (or diffusion-controlled) processes. Dividing $v^{0.5}$ on both sides of Equation (1) gives:

$$i v^{-0.5} = k_1 v^{0.5} + k_2 \quad (2)$$

Equation (2) shows that $i v^{-0.5}$ and $v^{0.5}$ are expected to have a linear relationship, with k_1 and k_2 being the slope and the y-intercept, respectively. Repeating the above step at other scan rates reveals the current density contribution across the potential window and outlines the contribution from the fast-kinetic and slow-kinetic processes. Figure 4c shows an example of the decoupling of a CV at 100 mV s⁻¹. The capacitive contribution from the fast-kinetic processes (yellow region) clearly dominates that of the slow-kinetic processes (blue region) at all scan rates (Figures 4c-d, S10)." [Page 16, revised main text]

Comment #6: *If possible, I suggest the authors test their best pCNF-MnO₂ electrodes with ionic liquid electrolyte and to expand the voltage window during CV experiments. The selection of the aqueous 6 M KOH limited the power density and energy density values of the prepared pseudo-capacitance electrode materials. By removing the aqueous electrolyte, a voltage window can be expanded to up to 3 V. It would be nice to compare the performance properties of these materials against other promising capacitor devices featuring MnO₂ - as demonstrated by Zhang et al. - J. Mater. Chem. A. 2013 DOI: 10.1039/C3TA00981E.*

Response: We appreciate the reviewer's kind suggestion on the use of ionic liquid electrolytes. Following the suggestion, we have prepared symmetric supercapacitors with two PCF@MnO₂-2h electrodes of an identical weight, a Celgard separator, and an ionic liquid electrolyte of 1-butyl-3-methylimidazolium tetrafluoroborate ([Bmim]BF₄) or 1-butyl-3-methylimidazolium hexafluorophosphate ([Bmim]PF₆). To prevent oxygen and moisture from contaminating the ionic liquid, the supercapacitors are encapsulated in LIR2032 button cases (Figure R3a) in a glove box filled with ultrapure argon. As the reviewer has suggested, the supercapacitors with neat ionic liquids indeed achieved a wide potential window of 3 V, however, their CVs substantially deviated from the ideal rectangular shape (Figure R3b, c). This distorted CV indicated the internal resistance of the supercapacitor was high, most likely due to the high

viscosity and resistance of the ionic liquids, as discussed in the paper provided by the reviewer [Zhang *et al.*, *J. Mater. Chem. A.* 2013, 1, 3706-3712].

Figure R3. (a) A photograph of a two-electrode symmetric supercapacitor encapsulated in a coin cell. (b, c) CV curves of a PCF@MnO₂-2h symmetric supercapacitor at 5 mV s⁻¹ with (b) 1-butyl-3-methylimidazolium tetrafluoroborate ([Bmim]BF₄) and (c) 1-butyl-3-methylimidazolium hexafluorophosphate ([Bmim]PF₆) ionic liquid electrolytes.

Following the work by Zhang *et al.*, we blended [Bmim]BF₄ and [Bmim]PF₆ with an equal volume of *N,N*-dimethylformamide (DMF) to reduce the electrolyte viscosity. The addition of DMF lowered the potential window to 2.0 V, as illustrated by the linear sweep voltammograms (Figure R4). This voltage window corroborated with the reports by Zhang *et al.* [*J. Mater. Chem. A.* 2013, 1, 3706-3712] and Chang *et al.* [*J. Mater. Chem.*, 2009, 19, 3732-3738]. With ionic liquid/DMF as the electrolytes, the capacitive performance of PCF@MnO₂ improved significantly (Figures R5 and R6). The CV shapes of the supercapacitors with DMF-blended ionic liquids (Figures R5a, b and R6a, b) became more rectangular than those of the supercapacitors with neat ionic liquids electrolyte (Figure R3b, c). Our preliminary tests showed that the maximal capacitances of MnO₂ of the supercapacitors with [Bmim]BF₄ and [Bmim]PF₆ electrolytes were 225.6 F g⁻¹ and 313.6 F g⁻¹, respectively, at 5 mV s⁻¹ (Figures R5c and R6c). These capacitances are higher than those of other reported MnO₂ electrodes in ionic liquids, including MnO₂-coated carbon nanotubes (~200 F g⁻¹, *J. Mater. Chem. A.* 2013, 1, 3706-3712], MnO₂ films (72 F g⁻¹, *J. Mater. Chem.*, 2009, 19, 3732-3738), and α -MnO₂ nanosheets (100 F g⁻¹, *J. Mater. Chem.*, 2012, 22, 6274-6279). The use of ionic liquid electrolytes enhanced the supercapacitor's energy density due to the larger voltage window than the KOH aqueous electrolyte (Figures R5d and R6d). However, the maximal capacitance is lower than the highest capacitance demonstrated by Zhang *et al.* (523.3 F g⁻¹), probably due to the lack of device optimization (*e.g.*, we have used symmetric capacitors instead of asymmetric capacitors in the report by Zhang *et al.*). We anticipate that tailoring the ionic liquid composition and the porous structure of PCFs can further improve the capacitive performance, as indicated by the different capacitances and energy densities between [Bmim]BF₄ and [Bmim]PF₆. A systematic study to elucidate the relationships among the ion diffusion, voltage window, capacitance, power and energy densities is undergoing. We will present our detailed findings in a future full paper.

We have revised the conclusion and added the paper suggested by the reviewer as Ref.60 in the manuscript:

"Future investigations on the interplays among the polymer molecular weight, mesopore size, mass loading of MnO_2 , ion diffusion resistivity and the use of ionic liquid electrolytes⁶⁰ are expected to further optimize the capacitive performance of PCF@MnO_2 and enhance the energy density of the supercapacitors." [Page 20, revised main text]

Figure R4. The linear scan voltammogram of a $\text{PCF@MnO}_2\text{-2h}$ symmetric supercapacitor at 5 mV s^{-1} with a mixed electrolyte containing equal volumes of DMF and (a) $[\text{BMIM}]\text{BF}_4$ or (b) $[\text{BMIM}]\text{PF}_6$. The blue dashed lines highlight the potential where the current starts to increase rapidly, indicating the decomposition of the electrolyte.

Figure R5. The capacitive performances of a PCF@MnO₂-2h symmetric supercapacitor with a mixed electrolyte containing equal volumes of [BMIM]BF₄ and DMF. (a and b) The cyclic voltammograms at (a) 5-80 mV s⁻¹ and (b) 100-1000 mV s⁻¹. (c) The rate capability performance. The capacitances are values normalized to the mass of MnO₂. (d) Histograms of the energy densities of the symmetric devices with aqueous KOH and [BMIM]BF₄/DMF electrolytes.

Figure R6. The capacitive performances of a PCF@MnO₂-2h symmetric supercapacitor with a mixed electrolyte containing equal volumes of [BMIM]PF₆ and DMF. (a and b) The cyclic voltammograms at (a) 5-80 mV s⁻¹ and (b) 100-1000 mV s⁻¹. (c) The rate capability performance. The capacitances are values normalized to the mass of MnO₂. (d) Histograms of the energy densities of the symmetric devices with aqueous KOH and [BMIM]PF₆/DMF electrolytes.

REVIEWERS' COMMENTS:

Reviewer #1 (Remarks to the Author):

The authors have adequately revised the manuscript to address all of my concerns. I believe that this work is now worthy of publication in Nature Communications.

Reviewer #2 (Remarks to the Author):

The manuscript has been suitably revised according to my comments and is now publishable as is.

Reviewer #3 (Remarks to the Author):

I am satisfied with the authors' responses and changes to the manuscript. I recommend the article for publication.